# Epidemiological trends of urolithiasis in working-age populations: Findings from the global burden of disease study 1990–2021

Weitao Yao[1], Xin Wei[2], Qiang Jing[2], Xiaobin Yuan[2], Fan liu[2], Xuhui Zhang[1]*

1 Department of Urology, The First Clinical Medical School of Shanxi Medical University, Taiyuan, Shanxi, China, 2 Department of Urology, The First Clinical Medical School of Shanxi Medical University, Taiyuan, Shanxi, China

☯ These authors contributed equally to this work.
¤ Department of Urology, The First Clinical Medical School of Shanxi Medical University, Taiyuan, Shanxi, China.
* xuhuizhangurology@126.com

## Abstract

### Background

One of the most prevalent urinary disorders worldwide, urolithiasis has symptoms and a high rate of recurrence. It has placed a significant strain on the global economy and health care system, particularly among those aged 20–54. Comprehensive research on the global burden and evolving trends of urinary calculi in the 20–54 age range is lacking.

### Methods

Three important urolithiasis indicators—prevalence, incidence, and Disability-Adjusted Life Years (DALYs)—were used from the Global Burden of Disease (GBD) database between 1990 and 2021 for this study.Point estimates were provided, as well as 95% uncertainty intervals (UIs).. To assess trends in the burden of urolithiasis among the 20–54 age range, we used percentage change and estimated annual percentage change (EAPC). The time trends from 1990 to 2021 were thoroughly examined using joinpoint regression analysis. This method makes it possible to compute the annual percent change (APC) and average annual percent change (AAPC), as well as the 95% Confidence Intervals (CIs)that go with them.

### Results

The number of patients, cases, and DALYs linked to urinary stones among people aged 20–54 has significantly increased worldwide. For example, the number of urolithiasis cases increased by 48%, from 1,576,704 in 1990–2,335,010 in 2021. Comparably, the number of new cases increased by 47%, from 42,272,855 in

**Data availability statement:** Data supporting the findings of this study can be found at the following location: Public Data Repository The data are stored in Global Burden of Disease (GBD), and can be accessed via the following URL: https://vizhub.healthdata.org/gbd-results/

**Funding:** The author(s) received no specific funding for this work.

**Competing interests:** The authors have declared that no competing interests exist.

1990–62,269,033 in 2021. Between 1990 and 2021, the number of DALYs rose from 218,150–290,210, representing a roughly 33% increase. As of 2021, intermediate sociodemographic index (SDI) regions had the highest DALY rates, while medium-high SDI regions had the highest prevalence and incidence rates among the five SDI regions examined for this time period. Individuals between the ages of 50–54 had the greatest incidence rate within the designated cohort for that year, followed by those between the ages of 45–49 and 40–44. Across all age groups, there were noticeable gender differences, with males showing significantly greater rates than girls.

## Conclusions

Overall, over the past 32 years, the prevalence of urolithiasis among people aged 20–54 has increased dramatically worldwide, especially in low-SDI nations and among those aged 50–54. With the goal of reducing the social and medical cost and enhancing the working-age population's quality of life and productivity, the study's findings highlight the urgent need for focused intervention techniques to prevent and treat urolithiasis in this age group. For example, in middle – high SDI areas, encourage 3 liters of water and a low – salt diet daily, especially for the over - 50s. In middle SDI areas, enhance primary care diagnostics, like using portable ultrasound. Regular screenings should be set for males in high – risk jobs, such as heat – exposed ones. However, this GBD – based study has limitations like data uncertainty, insufficient local strategies, and lack of trend tracking.

## Introduction

Urolithiasis is among the most prevalent urinary disorders globally, with prevalence estimates varying from 1% to 13% across different regions of the world [1]. The incidence ranges from 7 to 13% in North America, 5–9% in Europe, and 1–5% in Asia [2–4]. With the passage of time, the number of patients with urinary stones is also increasing year by year. Recent evidence suggests that the changes in the prevalence of urolithiasis may be attributable to a combination of factors, including geographical location, social conditions, dietary habits, climatic conditions, comorbidities, genetic variations, and the adverse side effects of certain medications [5–7]. Urolithiasis presents symptoms and a high recurrence rate that considerably diminish patients' quality of life, while also heightening their risk for comorbidities such as fractures, renal dysfunction, obesity, diabetes, and cardiovascular diseases [1]. It also places a heavy burden on the world's health care systems and the world economy [8,9]. One study showed that the estimated cost of treating urolithiasis in the United States was $3.79 billion in 2007 and is expected to increase by $1.24 billion per year by 2030 [10]. Urinary calculi have become a huge burden on public health.The incidence of stones varies by age, with lower incidence in children and the elderly [11,12]. Several studies have found that urolithiasis is more common in the working

age population, with a second peak of urolithiasis occurring between 40 and 49 years of age for each gender since 2000. This research reveals that calcium – containing stone formation is more prevalent among individuals aged 20–29, with male patients reaching a plateau between 30–69 years old at 19.6%. In older age groups, men are three times more likely to develop calcium – containing stones than women. Since 2000, middle – aged adults (40–49 years old) of both genders have experienced a significant increase in calcium – containing stones. Calcium stone prevalence is notably higher in individuals over 40 compared to their proportion in the general population. Uric acid stones typically become more common starting in middle age (30–39 years old), while cystine stones are more prevalent at younger ages. The highest incidence rates were observed in women aged 20–29 and men aged 30–39 [13]. It is evident that the incidence of urinary calculi is highest among those aged 20–54, who are also the primary producers of social and economic goods. Urinary calculi's acute pain, surgical necessity, and recurrence issues (which can occur up to 50% of the time) have a direct impact on productivity and labor force participation. The 20–54 age group is more receptive to health education than the elderly, and lifestyle changes like drinking water and nutrition (such as reducing salt and purine intake) can successfully lower the incidence rate. After the age of 60, the chance of developing chronic kidney disease increases fourfold if stones in patients aged 20–54 are not promptly removed, adding to the medical burden of an aging society.Therefore, it is necessary to comprehensively describe and analyze the overall incidence and changing trend of urinary stones in working age population. It is the strategic requirement for preserving the health of the workforce, maximizing the distribution of health resources, and achieving healthy aging, in addition to being the primary entry point for the prevention and control of disease burden.The Global Burden of Disease 2021 study is a systematic review of published and publicly available evidence on the incidence, prevalence and mortality of 369 diseases and injuries in 204 countries and territories and 21 regions for the period 1990–2021 [14]. Based on the latest data from the GBD2021 study, we analyzed the urinary stone incidence, prevalence and DALYs (disability-adjusted life years: It refers to all the years of healthy life lost from disease onset to death, including years of life lost due to premature death (YLL) and years lived with disability (YLD), the sum of YLL and YLD). for people aged 20–54 years at global, regional and country levels from 1990 to 2021, and compared the distribution and changes of urinary stone burden in different gender groups.

## Methods

### Data acquisition and download

A thorough assessment of the health hazards related to 369 illnesses and injuries, as well as 88 risk factors, in 204 nations and territories is provided by the 2021 Global Burden of Disease (GBD) research [15]. The most recent epidemiological data and improved standardized procedures are used in this study. The study used the GBD 2021 data to calculate prevalence, incidence, and disability-adjusted life years (DALYs) associated with urinary stones, as well as their 95% UI. All this datais accessible for free access through the Global Health Data Exchange (https://ghdx.healthdata.org/gbd-2021/sources)

### Socio-demographic index (SDI)

The Institute for Health Metrics and Evaluation (IHME) introduced the Socio-demographic Index (SDI) in 2015 as a comprehensive tool to assess a country's or region's level of development, emphasizing the relationship between social development and population health outcomes. It combines the geometric mean of lagged per capita income, scaled from 0 to 1, the average educational attainment of those aged 15 and over, and the overall fertility rate of people under 25. A scale from 0 to 100 was created for the GBD 2021 by multiplying the SDI values by 100. A score of 0 denotes low income and degree of education with high fertility, while a score of 100 denotes high income and degree of education with low fertility. Five SDI groups—low, medium-low, medium, medium-high, and high—were assigned to the 204 nations and territories in 2021 [15].

## Disability-adjusted life years

Disability-adjusted life years (DALYs), which represent the years of healthy life lost as a result of both premature mortality and disability, are a common metric used to evaluate health burden. DALYs = YLLs + YLDs is the formula that can be used to compute this. DALYs were equal to years lived with disability (YLDs) in this study because years of life lost (YLLs) were set to 0 because urinary stone-related fatalities could not be directly attributed in the GBD computation. To produce an estimate of the draw level, the computations were performed 500 times. Since simulation testing showed that this change had no discernible impact on the final estimates or their uncertainties, the number of computations was lowered from 1000 in earlier GBD iterations to 500 for GBD 2021. The 2.5th and 97.5th percentiles represent the 95% uncertainty interval, while the final estimate is based on the average of the 500 draws. Every stage of the estimating process takes uncertainty into consideration [15].

## Estimated annual percentage change and percentage change

In epidemiology, public health, and statistics, the term "estimated annual percentage change" (EAPC) is frequently used to describe the average yearly percentage change of an index over time, such as disease incidence, mortality, etc. It is frequently used to examine the rising or downward patterns of long-term trends and uses statistical models to predict the degree and direction of trend changes. EAPC > 0 indicates an annual average increase in the indicator, whereas EAPC < 0 indicates an annual average decrease. A reliable and popular metric for tracking changes in indicators like prevalence and incidence rates over predetermined time periods is the estimated annual percent change (EAPC) [16]. Estimating the dynamic trends in urinary stone incidence, prevalence, and DALYs between 1990 and 2021 was the goal of this study. By fitting the natural log of each observation to a straight line with time as a variable and calculating the slope of this line, the EAPC was determined using the natural log rate of the fitted regression model [17].

$$y = \alpha + \beta x + \varepsilon$$

$$\text{EAPC} = 100 \times (\exp(\beta) - 1)$$

The year is represented by X in the model, the natural logarithm of rates (such prevalence and incidence) by y, the intercept by $\alpha$, the slope by $\beta$, and random error by $\varepsilon$. This fitted model was used to calculate the estimated annual percentage change's (EAPC) 95% CI. The 95% confidence intervals were used to interpret the trend results: if the lower limit of the 95% CI was more than 0, a trend was considered upward; if the upper limit was less than 0, a trend was considered downward. It would indicate that the trend change was not statistically significant if the 95% CI contained 0 [18]. Additionally, this study used percentage changes to show how prevalence, incidence, and DALYs changed between 1990 and 2021.

$$\text{Percentage change} = (2021 \text{ cases} - 1990 \text{ cases}) / 1990 \text{ cases}$$

Data cleaning, calculations, and plotting were performed with the use of R software, version 4.4.1. Visualizations were created using the ggplot2 package.

## Joinpoint regression analysis

In order to evaluate temporal trends in disease frequency or death, we employed a connective-point regression model, a statistical technique frequently used in epidemiologic investigations [19]. One statistical technique for identifying turning points ("joining points") of trends in time-series data is joinpoint regression analysis. By splitting time series into several linear intervals and determining the points at which trends change considerably, it is frequently used in epidemiology,

public health, and economics to examine long-term trend changes in morbidity, mortality, or economic indicators.Important shift points in time-series data on the prevalence of urinary stones from ages 20–54 were statistically defined and expertly detected by the model using global, continental, and national scale data. To illustrate changes in prevalence throughout the period shown, the model makes it easier to compute the annual percent change (APC) and the 95% confidence interval (CI) that goes with it. Additionally, the average annual percent change (AAPC), which takes into account aggregated trend data for the study period from 1990 to 2021, was computed in order to thoroughly evaluate the patterns that were detected. An upward trajectory across the designated interval is shown statistically by an APC or AAPC estimate where the lower bound of its 95% CI is greater than zero. On the other hand, a decreasing trend is indicated by an APC or AAPC estimate plus a 95% CI upper limit below zero. The trend is said to be stable when the 95% CI of APC or AAPC is zero.

### Statistics analysis

Prevalence, incidence, mortality, and disability-adjusted life years are expressed as predictions per 100 000 population, including their 95% UI. All analyses and graphical presentation procedures were performed with the use of statistical software R, version 4.4.1.

## Results

### Global level

Globally, the number of patients, the number of cases, and the number of DALYs associated with urinary stones in the 20−54 age group increased significantly.For example, urolithiasis cases increased from 1.58 million in 1990 to **2.34 million** in 2021, a percentage change of 48%; The number of new cases increased from **42.27 million** in 1990 to **62.27 million** in 2021, with a percentage change of 47%. DALYs cases increased from **0.22 million** in 1990 to **0.29 million** in 2021, a percentage change of 33% (Table 1 and S1 and S2). This indicates that the burden of urinary stones among persons 20–54 years of age continues to increase worldwide (Table 1 and S1–S3 and Figs 1−2 and S1–S3). However, from 1990 to 2021, the global prevalence, incidence and DALY rate of urinary stones showed a downward trend among the 20−54 age group, and the EAPC was −0.17 (95%CI:-0.2--0.14) and −0.19 (95%CI:-0.22--0.15), respectively. −0.6 (95%CI:-0.67--0.54) (Table 1 and S1 and S2, Fig 1A~C). While the age-standardized rate shows a downward trend (1990–2021), this suggests that the disease burden of urinary calculi in people aged 20–54 years worldwide is still increasing. This suggests that while aging and population growth may be the primary causes of the absolute increase, the age-standardized risk is actually declining. This could imply that aging and population growth worldwide, rather than elevated personal risk, are the causes of the growing burden of disease.

### SDI regional level

In 2021, the prevalence, incidence and absolute number of DALYs of urinary stones in 20–54 years old group were the highest in middle SDI **0.82 million** (95%UI: 0.63 **million**-1.06 **million**), **21.87 million** (95%UI:16.64 **million**-28.10 **million**) and **0.10 million** (95%UI:0.08 **million**-0.14 **million**), respectively. In 2021, the highest prevalence, incidence, and DALYs rates were found in moderately high SDI regions, while the highest DALYs rates were found in moderate SDI regions (Table 1 and S1, S2). The prevalence rate and incidence rate increased with the increase of SDI, but reached the highest level in the middle and high SDI areas. DALYs increased gradually with the increase of SDI, and reached the highest level in the middle SDI area and then decreased, with the lowest SDI area having the largest percentage of change (about 150%). In contrast, regions with high SDI showed the smallest percentage change (about −4% to −5%) (Table 1 and S1, S2, Fig 2A and 2B). It is noteworthy that from 1990 to 2021, the prevalence and incidence of SDI in the central and low-middle regions showed a rapid increase, with an EAPC of 0.37 (95%CI:0.34–0.39), respectively. 0.38 (95% CI: 0.3 0.46) and 0.37 (95% CI: 0.34 0.39), 0.38 (95% CI: 0.3 0.46) (Table 1 and S1, S2, Fig 2A and 2B). Therefore,

**Table 1. The prevalence of urinary stones and rates among 20-54years in 1990 and 2021, and the trends from 1990 to 2021.**

| location | Prevalent cases | | | Prevalent rates | | |
|---|---|---|---|---|---|---|
| | 1990 (95% UI) | 2021 (95% UI) | percentage change (100%) | 1990_per 100 000(95% UI) | 2021_per 100 000(95% UI) | EAPC(95% CI) |
| Andean Latin America | 13641.64 (10340.71-17743.4) | 32130.35 (25316.93-40396.63) | 1.36 | 86.99 (65.94-113.15) | 98.7 (77.77-124.09) | 0.55 (0.47-0.63) |
| Australasia | 6632.33 (4812.97-8830.34) | 9712.37 (7094.43-12938.02) | 0.46 | 65.9 (47.83-87.75) | 66.59 (48.64-88.71) | 0.01 (−0.05-0.06) |
| Caribbean | 6980.96 (5227.01-9169.39) | 12941.37 (9628.09-17067.11) | 0.85 | 43.95 (32.91-57.73) | 56.42 (41.97-74.41) | 0.88 (0.82-0.94) |
| Central Asia | 23099.71 (17806.24-29895.51) | 38566.3 (29738.82-50010.27) | 0.67 | 77.69 (59.89-100.54) | 82.7 (63.77-107.24) | 0.28 (0.22-0.33) |
| Central Europe | 37361.34 (28545.98-49353.64) | 28177.67 (21630.76-36450.55) | −0.25 | 63 (48.14-83.22) | 51.53 (39.56-66.66) | −0.65 (−1.06--0.23) |
| Central Latin America | 30951.3 (22949.85-41023.79) | 74020.83 (56836.28-95923.81) | 1.39 | 45.38 (33.65-60.15) | 59.24 (45.49-76.77) | 1.06 (0.75-1.37) |
| Central Sub-Saharan Africa | 5609.15 (4214.43-7390.99) | 16435.49 (12342.21-21681.74) | 1.93 | 27.75 (20.85-36.56) | 30.24 (22.71-39.9) | 0.28 (0.24-0.31) |
| East Asia | 363629.48 (272911.59-477666.61) | 398221.63 (300760.52-522276.37) | 0.1 | 59.77 (44.86-78.52) | 54.13 (40.88-70.99) | −0.54 (−0.72--0.37) |
| Eastern Europe | 218017.52 (169446.46-280782.63) | 181003.68 (137307.79-234287.09) | −0.17 | 197.62 (153.59-254.51) | 183.72 (139.37-237.81) | −0.31 (−0.38--0.23) |
| Eastern Sub-Saharan Africa | 24253.64 (18450.39-31857.76) | 61368.88 (46995.15-80034.29) | 1.53 | 35.79 (27.23-47.01) | 35.79 (27.41-46.67) | −0.13 (−0.16--0.1) |
| Global | 1576704.02 (1189918.14-2050374.76) | 2335010.67 (1789652.77-3011581.44) | 0.48 | 65.6 (49.51-85.3) | 61.94 (47.48-79.89) | −0.17 (−0.2--0.14) |
| High-income Asia Pacific | 69092.75 (48702.04-93392.12) | 71267.73 (51908.56-95627.11) | 0.03 | 78.44 (55.29-106.02) | 84.69 (61.68-113.63) | 0.24 (0.2-0.29) |
| High-income North America | 120490.75 (87976.31-161646.63) | 75780.83 (61277.52-95897.81) | −0.37 | 85 (62.06-114.04) | 45.09 (36.46-57.05) | −2.36 (−2.66--2.07) |
| High-middle SDI | 452912.52 (344305.36-593687.43) | 496179.72 (376310.69-646085.54) | 0.1 | 86.96 (66.11-113.99) | 75.87 (57.54-98.79) | −0.56 (−0.65--0.46) |
| High SDI | 333435.64 (243820.41-444243.36) | 319561.91 (242667.53-420797.41) | −0.04 | 75.48 (55.19-100.56) | 61.89 (47-81.5) | −0.65 (−0.71--0.6) |
| Low-middle SDI | 242287.73 (183342.15-315918.1) | 512644.57 (389957.46-665919.08) | 1.12 | 51.6 (39.04-67.28) | 55.99 (42.59-72.73) | 0.38 (0.3-0.46) |
| Low SDI | 74907.63 (56533.66-97749.23) | 186352.38 (140638.96-243007.81) | 1.49 | 40.62 (30.66-53) | 41.31 (31.18-53.87) | 0.13 (0.06-0.2) |
| Middle SDI | 471893.93 (357924.19-613987.94) | 818595.94 (628062.05-1056541.33) | 0.73 | 60.13 (45.61-78.24) | 66.58 (51.08-85.93) | 0.37 (0.34-0.39) |
| North Africa and Middle East | 54554.24 (40120.64-71925.99) | 147156.65 (106628.63-195819.23) | 1.7 | 40.66 (29.91-53.61) | 47.43 (34.37-63.11) | 0.5 (0.49-0.51) |
| Oceania | 1102.49 (818.78-1474.27) | 2820.19 (2086.91-3657.22) | 1.56 | 40.8 (30.3-54.55) | 44.72 (33.09-57.99) | 0.31 (0.24-0.37) |
| South Asia | 258493.82 (194493.53-336950.14) | 593809.14 (448009.09-771962.57) | 1.3 | 56.68 (42.65-73.89) | 64.91 (48.97-84.39) | 0.7 (0.54-0.85) |
| Southeast Asia | 138294.46 (105152.16-176866.94) | 266248.39 (203143.28-342870.21) | 0.93 | 68.04 (51.73-87.01) | 75.11 (57.31-96.73) | 0.29 (0.26-0.32) |
| Southern Latin America | 21694.05 (15620.03-28813.03) | 38855.37 (30241.37-49518.84) | 0.79 | 97.56 (70.24-129.57) | 116.09 (90.36-147.96) | 0.3 (0.19-0.41) |
| Southern Sub-Saharan Africa | 7706.3 (5763.48-10127.17) | 14404.59 (10683.98-18953.65) | 0.87 | 35.79 (26.77-47.03) | 36.65 (27.18-48.22) | 0.04 (−0.02-0.09) |
| Tropical Latin America | 19129.98 (14804.06-24669.7) | 62801.62 (49747.73-80689.76) | 2.28 | 28.07 (21.72-36.2) | 53.84 (42.65-69.18) | 1.72 (1.22-2.22) |

*(Continued)*

**Table 1.** (Continued)

| location | Prevalent cases | | | Prevalent rates | | |
|---|---|---|---|---|---|---|
| | 1990 (95% UI) | 2021 (95% UI) | percentage change (100%) | 1990_per 100 000(95% UI) | 2021_per 100 000(95% UI) | EAPC(95% CI) |
| Western Europe | 134627.5 (99307.53-178619.17) | 153819.66 (117555.7-202417.95) | 0.14 | 71.25 (52.56-94.53) | 78.26 (59.81-102.99) | 0.6 (0.38-0.81) |
| Western Sub-Saharan Africa | 21340.62 (15960.79-28171.57) | 55467.94 (41430.14-72713.77) | 1.6 | 29.96 (22.41-39.55) | 29.33 (21.91-38.45) | −0.1 (−0.14--0.05) |

the prevalence and incidence of urinary calculi in the 20–54 age group were the highest in the middle and high SDI regions, but the rate of increase was the fastest in the middle SDI region. Incidence and prevalence rise with SDI, peaking in moderate to high SDI regions. DALY rates peak in moderate SDI areas, then decline. Low-SDI regions see the largest increase (+150%), while high-SDI areas show minimal change due to better prevention. Moderate-SDI regions carry the heaviest and fastest-growing disease burden.

## GBD regional level

The prevalence, incidence, and absolute numbers of DALYs of urinary stones in the 20–54 age group increased over time and were observed in most regions. Only a few regions, notably East Asia, Eastern Europe, and high-income North America, saw a decline, all of which are high SDI/ moderate-to-high SDI regions. Over the past 31 years, prevalence and disability-adjusted life years (DALYs) have continued to increase in most regions, with the greatest increases in Central Asia and southern sub-Saharan Africa; The EAPC for prevalence was 0.28 (95%CI: 0.22–0.33) and 0.04 (95%CI:-0.02–0.09), respectively, and the EAPC for DALYs was −0.37 (95%CI:-0.02–0.09), respectively. −0.53--0.21) and 0.06 (95%CI:-0.35–0.46). In contrast, in high-income North America, East Asia, and Eastern Europe, both prevalence and DALY rates have declined; The EAPC for prevalence was −2.36 (95% CI: 2.66–2.07), 0.54 (95% CI: 0.72–0.37) and 0.31 (95% CI: 0.38–0.23), the DALY EAPC respectively 1.03 (95% CI: 1.28–0.78), 1.95 (95% CI: 2.17–1.72) and 0.74 (95% CI: 0.87–0.61). The incidence and prevalence of urinary calculi in different regions showed the same trend in 20–54 age group. In East Asia, Eastern Europe, and high-income North America, there is a clear downward trend in the prevalence, incidence, and number and rate of DALYs of urinary stones in the 20–54 age group, indicating that the burden of urinary stones is decreasing in this region. This trend is likely to be closely related to improvements in health care and economic conditions (Tables 1, S1 and S2, Fig 3A and 3B). In addition to having the highest disease burden, the areas with moderate SDI are also expanding the fastest, necessitating priority action. While low SDI locations required resources to contain the danger, high SDI areas could successfully lower the burden through public health expenditure.

## National level

From 1990 to 2021, the prevalence, incidence and DALYs of urinary stones among 20–54 years old have increased in about 86% of countries. The U.S. Virgin Islands, Trinidad and Tobago, and Brazil had the highest percentage change, with all three classified as moderate SDI regions, ranging from 80% to 95%. This is in sharp contrast to the significantly lower percentage changes observed in other regions with moderate SDI. In addition, only 13% of countries showed a reduction in prevalence, incidence and DALYs, such as the United States (high SDI), Poland and Australia (high SDI), with percentage reductions ranging from 39% to 41%. This suggests that the increase in urinary stone cases in the 20–54 age group is polarized between countries in high SDI regions and those in middle SDI. Over the past 32 years, prevalence rates and disability-adjusted annual rates have increased in most countries; The largest increase observed from high SDI regions was in Spain, with an EAPC of 1.9 (95% CI: 1.63–2.16), contrary to the global trend in high SDI regions. In some countries with low SDI, the incidence showed an increasing trend. Most significant increase was seen in Syrian Arab Republic,

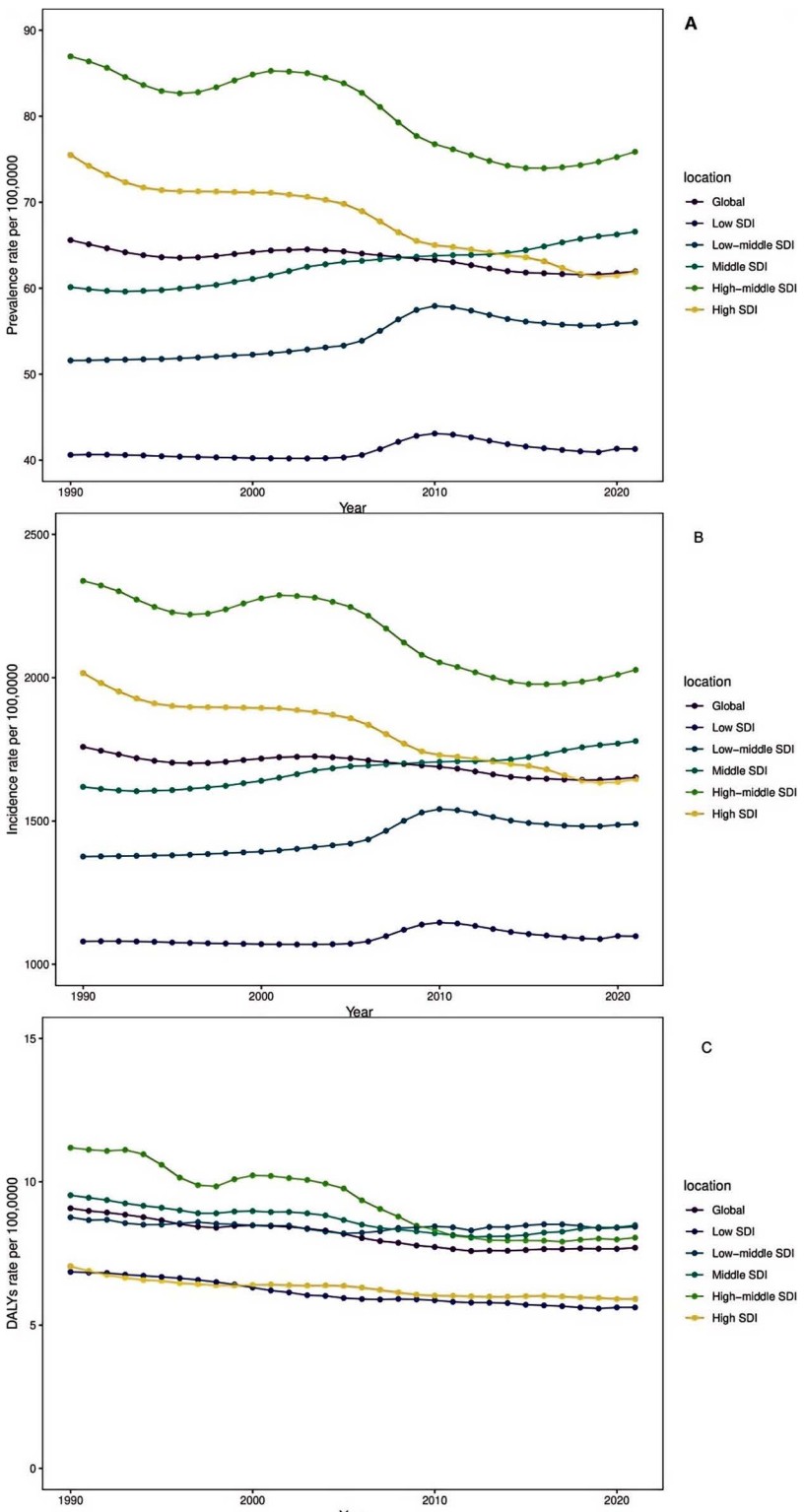

**Fig 1. Temporal trend of migraine burden in WCBA in global and 5 territories.** (A)~(C) The rates of prevalence, incidence, and DALYs from 1990 to 2021. DALYs = Disability-Adjusted Life Years.

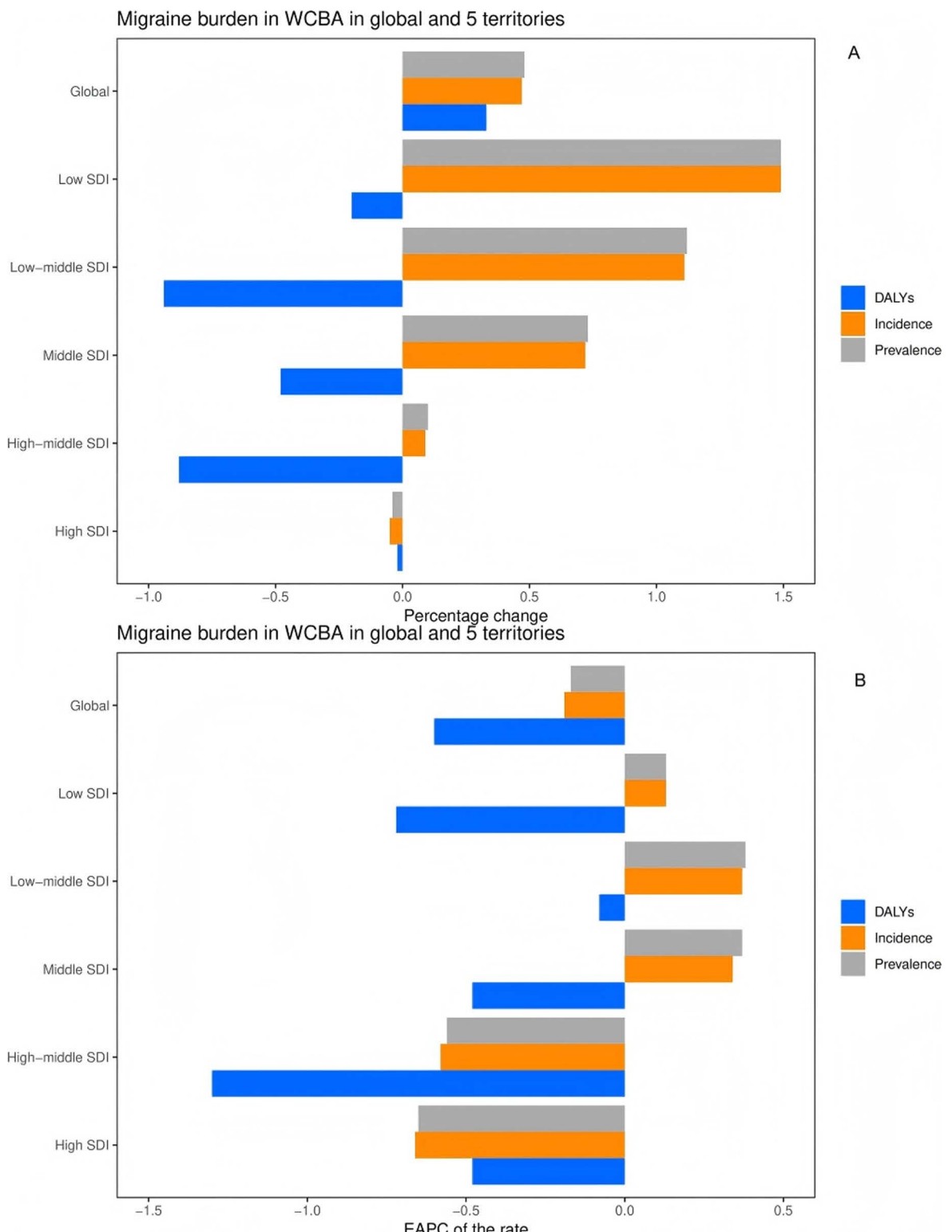

**Fig 2. Temporal trend of migraine burden in 20-54years in global and 5 territories.** (A) Percentage change in cases of prevalent, incident, and DALYs in 1990 and 2021. (B) The EAPC of prevalence, incidence, and DALY rates from 1990 to 2021. EAPC = Estimated Annual Percentage Change, DALYs = Disability-Adjusted Life Years.

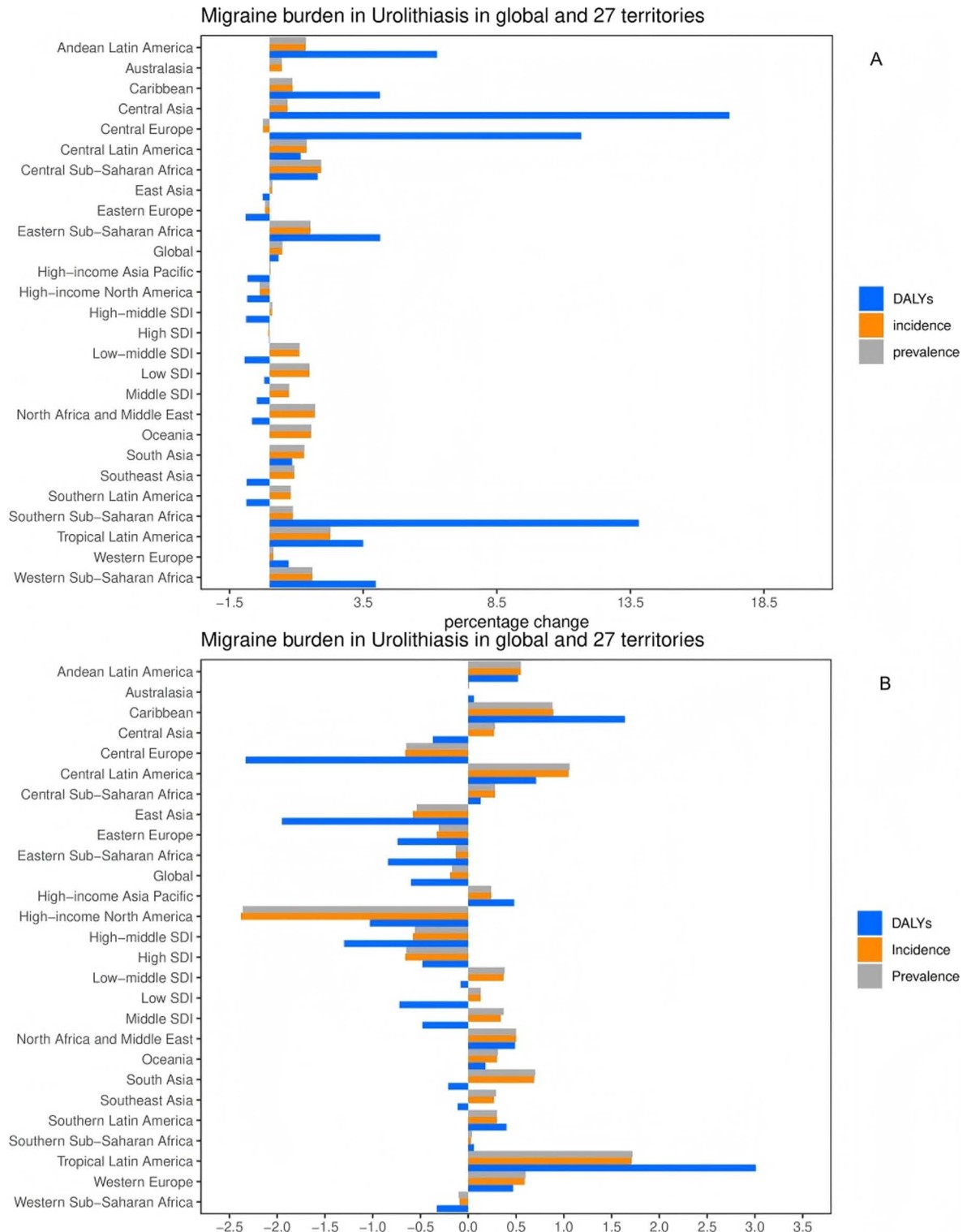

**Fig 3. Temporal trend of migraine burden in WCBA in regions.** (A) Percentage change in cases of prevalent, incident, and DALYs in 1990 and 2021. (B) EAPC of rates of prevalent, incident, and DALYs from 1990 to 2021. EAPC = Estimated Annual Percentage Change, DALYs = Disability-Adjusted Life Years.

EAPC = 0.87 (95%CI: 0.74–1.01), respectively. A few countries showed decreases in incidence, with the most pronounced reductions in the United States, with Eapcs of − 2.6 (95% CI, − 2.92 to 2.27), and in Poland and Australia, with Eapcs of − 2.46 (95% CI, − 3.26 to 1.66) (Table 2 and S3, Fig 4A ~ D). More attention has to be paid to some countries' abnormally high growth rates. Effective prevention and control led to a decrease in the burden of disease in nations with high SDI, which may highlight the critical role that economic investment and medical intervention play in reducing disease.

**Age and sex patterns**

In 2021, the global prevalence of Urolithiasis peaked in the 50–54 age bracket within the 20–54 age range and subsequently declined with advancing age. Across all ages, men exhibited a higher prevalence than women (Fig 5A; Supplementary S4 table). Regarding new Urolithiasis cases in 2021, the incidence was notably higher among men (2178 cases) compared to women (1115 cases) (Supplementary S8 Table). Specifically, the highest incidence occurred in the 50–54 age group, closely followed by the 45–49 and 40–44 age groups. Notably, significant gender disparities were observed in each age group, with men consistently having higher incidence rates than women. It is worth mentioning that the incidence rate among men aged 20–24 increased more rapidly than that among women in the same age group, and the largest gender gap was observed in the 50–54 age group (4118 men vs. 1910 women) (Fig 5B; Supplementary S5 Table). Analysis of Disability-Adjusted Life Years (DALYs) rates revealed similar trends to prevalence and incidence, with DALYs rates escalating with age. However, the sex-related DALYs rate was most comparable in the 20–24 age group (men: 2.78, women: 2.54), whereas the widest sex-related DALYs rate disparities were found in the 50–54 age group (men: 19.53, women: 11.67) and the 45–49 age group (men: 15.15, women: 9.16) (Fig 5C; Supplementary S6 Table). Interestingly, in every age group, men had greater disease burdens than women. This cohort revealed the narrowest gender disparity in DALY rates (male 2.78 vs female 2.54), indicating that gerden-related risk factors may worsen with age, even though males aged 20–24 years showed faster incidence rise than females of the same age. Men between the ages of 50 and 54 had the highest burden of urolithiasis, which calls for focused preventative measures.

**Overall temporal trends in gender and age structures**

From 1990 to 2021, the prevalence and incidence of urolithiasis within all age groups ranging from 20 to 54 years old generally fluctuated insignificantly. Specifically, throughout this period, the prevalence and incidence rates among males were consistently much higher than those among females. Despite a notable increase being observed during the period from 1995 to 2005, and a distinct downward trend emerging from 2005 to 2021, the overall prevalence and incidence ultimately demonstrated a downward trend (Fig 6A-B; Supplementary S7 and S8 tables). Conversely, the number of individuals affected and the number of cases in the 20–54 age group increased significantly, with the most pronounced increase occurring in the interim from 2000 to 2005 (Fig 6C-D). Among the 20–54 age group, the overall variations in DALYs were similar to those in prevalence. It is notable that DALYs for males remained higher than those for females (Fig 6E; Supplementary S9 Table). Gender differences in the burden of urolithiasis have existed for a long time, and the contradiction between the increase in the absolute number of urolithiasis cases and the decrease in the age-standardized rate is mainly due to the change in population structure

**Temporal joinpoint analysis**

Joint point regression analysis revealed that from 1990 to 2021, the prevalence of urolithiasis among individuals aged 20–54 years exhibited a significant overall downward trend (AAPC = −18.39%; 95% CI: −21.01% to −15.77%). Notably, there was an upward trend observed from 1995 to 2003 (APC = 23.2%; 95% CI: 20.34%−26.06%; P < 0.001) (Fig 7A and Supplementary S10 Table). Similarly, the incidence of urolithiasis in this age group also demonstrated a marked overall decline (AAPC = −20.11%; 95% CI: −22.74% to −17.48%; P < 0.001), with the most pronounced reductions occurring between 1990–1995 and during the period of 2011–2014 (APC = −68.03%; 95% CI: −72.88% to −63.19%; P < 0.001 and

**Table 2. Percentage change in prevalent, incidence, DALYs rate across 204 countries in 1990 and 2021.**

| location | prevalence rate (100%) | | | incidence rate (100%) | | | DALYs rate (100%) | | |
|---|---|---|---|---|---|---|---|---|---|
| | 1990 | 2021 | Percentage change | 1990 | 2021 | Percentage change | 1990 | 2021 | Percentage change |
| Afghanistan | 40.21854348 | 39.8225766 | −0.009845381 | 1065.941801 | 1060.985663 | −0.004649539 | 4.445796529 | 6.590686941 | 0.482453571 |
| Albania | 52.48388355 | 53.14677275 | 0.012630338 | 1387.147702 | 1403.871074 | 0.012055942 | 5.160752753 | 4.33862807 | −0.159303249 |
| Algeria | 38.3688206 | 46.69002339 | 0.216874083 | 1023.353597 | 1240.546186 | 0.212236112 | 3.229749441 | 4.520367918 | 0.399603282 |
| American Samoa | 41.90513563 | 54.67702436 | 0.30478099 | 1126.534374 | 1462.24029 | 0.297998822 | 3.596223413 | 4.364169019 | 0.213542241 |
| Andorra | 66.65258289 | 78.42613416 | 0.176640586 | 1780.627608 | 2083.343466 | 0.170005147 | 6.201660506 | 6.762393999 | 0.09041667 |
| Angola | 28.89391501 | 29.76405454 | 0.030114975 | 770.3481421 | 794.0717207 | 0.030795918 | 3.64859147 | 3.391958187 | −0.070337632 |
| Antigua and Barbuda | 41.99886307 | 54.42137273 | 0.295782046 | 1125.029148 | 1457.0599 | 0.2951308 | 3.967588698 | 6.28499267 | 0.584083721 |
| Argentina | 99.57139201 | 99.36289742 | −0.002093921 | 2648.352821 | 2640.918938 | −0.002806984 | 7.584359637 | 7.771430148 | 0.024665301 |
| Armenia | 87.65190982 | 116.6087306 | 0.33036155 | 2331.953466 | 3109.606768 | 0.333477196 | 14.16456739 | 12.72564964 | −0.101585718 |
| Australia | 64.8522807 | 66.45198814 | 0.024666942 | 1733.300088 | 1771.34853 | 0.021951445 | 6.20139139 | 6.138312302 | −0.010171764 |
| Austria | 151.1525015 | 104.0235889 | −0.311797106 | 4066.943343 | 2775.677002 | −0.317502909 | 13.16084724 | 9.077860991 | −0.310237341 |
| Azerbaijan | 71.0241219 | 77.56776442 | 0.092132678 | 1884.031748 | 2054.295145 | 0.09037183 | 6.534869405 | 6.46090299 | −0.011318729 |
| Bahamas | 41.93943245 | 55.14324287 | 0.314830451 | 1123.83068 | 1476.117859 | 0.313469979 | 4.754441308 | 8.550542814 | 0.79843272 |
| Bahrain | 45.02884983 | 56.25575804 | 0.249327004 | 1201.501747 | 1493.802521 | 0.243279524 | 3.575767067 | 4.986230259 | 0.394450524 |
| Bangladesh | 49.88411315 | 55.44515986 | 0.111479314 | 1326.705029 | 1472.960049 | 0.110239289 | 8.053960568 | 6.100234562 | −0.242579535 |
| Barbados | 44.93584853 | 62.92962598 | 0.400432573 | 1204.106026 | 1688.258126 | 0.402084276 | 5.585560885 | 9.712389574 | 0.738838726 |
| Belarus | 166.0462758 | 182.3806738 | 0.098372564 | 4451.285077 | 4879.791432 | 0.096265763 | 20.7916451 | 22.26221231 | 0.070728757 |
| Belgium | 72.49945897 | 75.22121826 | 0.037541788 | 1934.149303 | 2002.597865 | 0.035389492 | 6.343763184 | 6.385515463 | 0.006581626 |
| Belize | 39.83578045 | 51.56142227 | 0.294349494 | 1068.326178 | 1385.250701 | 0.296655206 | 4.082122915 | 8.527155591 | 1.088902213 |
| Benin | 27.55177608 | 28.5974855 | 0.037954338 | 732.1005308 | 760.3143464 | 0.038538171 | 4.00297093 | 3.63551403 | −0.091796045 |
| Bermuda | 47.24590934 | 62.71738116 | 0.327466907 | 1263.302073 | 1676.082287 | 0.326747041 | 4.38862683 | 6.668462165 | 0.519487171 |
| Bhutan | 48.70817509 | 55.22780635 | 0.133850863 | 1297.070722 | 1467.805659 | 0.13163117 | 8.344327524 | 6.783781893 | −0.187018741 |
| Bolivia (Plurinational State of) | 83.5861334 | 86.99020688 | 0.040725337 | 2231.949654 | 2318.894291 | 0.038954569 | 8.944007539 | 8.598186315 | −0.038665131 |
| Bosnia and Herzegovina | 53.67663816 | 54.6821663 | 0.018733069 | 1418.474907 | 1443.592629 | 0.017707555 | 6.265635816 | 4.933624897 | −0.212589904 |
| Botswana | 32.73826021 | 36.45486392 | 0.113524777 | 868.6096686 | 965.4781468 | 0.111521299 | 3.716714066 | 3.842875001 | 0.033944213 |
| Brazil | 27.78205622 | 54.23390643 | 0.952119959 | 742.2219183 | 1444.202306 | 0.945782347 | 4.465400096 | 11.88526074 | 1.661634004 |
| Brunei Darussalam | 61.3323749 | 69.14241378 | 0.12733958 | 1648.593543 | 1850.93616 | 0.122736509 | 6.262773557 | 6.902402894 | 0.10213196 |
| Bulgaria | 72.79724754 | 56.40680855 | −0.225151905 | 1942.004279 | 1488.835885 | −0.233350873 | 15.95803259 | 6.354863559 | −0.601776502 |
| Burkina Faso | 28.7589881 | 29.33207207 | 0.019927126 | 763.2646399 | 779.1513551 | 0.020814164 | 4.188461607 | 3.862127196 | −0.077912714 |
| Burundi | 34.24139513 | 34.84785431 | 0.017711287 | 909.2322915 | 926.2633541 | 0.018731256 | 6.113476418 | 4.947481307 | −0.19072538 |
| Cabo Verde | 26.3096341 | 31.63202737 | 0.202298263 | 699.9649311 | 839.0055261 | 0.198639373 | 2.74976423 | 3.144513042 | 0.143557331 |
| Cambodia | 51.38106252 | 61.03504705 | 0.187889935 | 1384.337856 | 1642.702993 | 0.186634452 | 9.344806083 | 9.133369791 | −0.022626076 |
| Cameroon | 28.7581961 | 29.17805092 | 0.014599484 | 763.4179243 | 775.4351796 | 0.015741385 | 5.243936458 | 4.404536074 | −0.160070663 |
| Canada | 45.87903576 | 48.71722405 | 0.061862422 | 1216.135806 | 1291.206889 | 0.061729194 | 4.305696835 | 5.015670728 | 0.164891752 |
| Central African Republic | 28.23181244 | 31.26594845 | 0.107472236 | 752.8927388 | 836.239588 | 0.110702156 | 4.018755739 | 4.209189786 | 0.04738632 |
| Chad | 28.21669313 | 28.0106903 | −0.007300743 | 749.4748528 | 744.826716 | −0.006201858 | 3.629786451 | 3.684612799 | 0.015104566 |
| Chile | 92.24935254 | 158.2475282 | 0.715432399 | 2459.965358 | 4224.29739 | 0.717218243 | 7.759731339 | 13.19288866 | 0.70017338 |
| China | 60.12042144 | 53.61598045 | −0.108190209 | 1627.179123 | 1437.364085 | −0.116652823 | 9.982153828 | 5.946545966 | −0.404282275 |
| Colombia | 37.82086222 | 43.00921194 | 0.137182217 | 1012.676014 | 1147.487222 | 0.13312373 | 4.912519211 | 4.913387738 | 0.000176799 |

*(Continued)*

**Table 2.** (Continued)

| location | prevalence rate (100%) | | | incidence rate (100%) | | | DALYs rate (100%) | | |
|---|---|---|---|---|---|---|---|---|---|
| | 1990 | 2021 | Percentage change | 1990 | 2021 | Percentage change | 1990 | 2021 | Percentage change |
| Comoros | 34.64357333 | 39.63379876 | 0.144044767 | 919.3405934 | 1049.64788 | 0.141739947 | 5.77326797 | 6.12303157 | 0.060583296 |
| Congo | 28.05466469 | 32.30964238 | 0.151667387 | 748.2190422 | 861.3627513 | 0.151217361 | 4.422164707 | 4.415134659 | −0.00158973 |
| Cook Islands | 44.30340303 | 53.68524314 | 0.211763419 | 1186.563236 | 1433.31756 | 0.207957163 | 3.586357669 | 4.072324351 | 0.135504243 |
| Costa Rica | 37.73233762 | 41.83833728 | 0.108819117 | 1010.871022 | 1116.128475 | 0.104125503 | 3.410504243 | 3.848061434 | 0.12829692 |
| Côte d'Ivoire | 28.22780646 | 29.92431282 | 0.060100538 | 749.9985858 | 794.8264102 | 0.059770545 | 3.891128622 | 3.928952569 | 0.009720559 |
| Croatia | 56.56291275 | 66.45234128 | 0.174839449 | 1494.402956 | 1765.110103 | 0.181147358 | 6.877628183 | 5.258944335 | −0.235354952 |
| Cuba | 46.38729908 | 73.44822508 | 0.583369296 | 1242.646308 | 1975.851974 | 0.590035685 | 5.358954035 | 10.59143465 | 0.976399608 |
| Cyprus | 55.88514059 | 57.9434711 | 0.036831445 | 1490.640448 | 1542.127802 | 0.034540424 | 5.639109367 | 5.072702632 | −0.100442587 |
| Czechia | 65.03519337 | 59.71163364 | −0.081856599 | 1729.479016 | 1576.928941 | −0.088205797 | 13.16744239 | 5.472970014 | −0.584355879 |
| Democratic People's Republic of Korea | 51.99568178 | 62.37636802 | 0.199645161 | 1399.969945 | 1680.381625 | 0.200298357 | 9.784420467 | 9.965706167 | 0.018527996 |
| Democratic Republic of the Congo | 27.33933776 | 30.17980614 | 0.103896752 | 728.3008079 | 805.1554966 | 0.105526024 | 3.115805388 | 3.382176554 | 0.085490309 |
| Denmark | 60.46205492 | 61.76826709 | 0.021603834 | 1610.860515 | 1644.672397 | 0.02098995 | 6.47975189 | 5.917751553 | −0.086731768 |
| Djibouti | 33.8049599 | 39.70632695 | 0.174571041 | 898.2596876 | 1053.892231 | 0.173260078 | 6.041138466 | 7.3061221 | 0.209394908 |
| Dominica | 42.13533579 | 51.74543638 | 0.228076991 | 1128.150965 | 1384.725866 | 0.227429581 | 3.460635555 | 4.749746215 | 0.372506911 |
| Dominican Republic | 40.0912517 | 47.36628168 | 0.181461782 | 1074.591564 | 1267.957724 | 0.179943865 | 3.776885432 | 4.602051405 | 0.218477894 |
| Ecuador | 91.89417255 | 119.3484548 | 0.298759774 | 2455.918274 | 3190.230172 | 0.298996879 | 7.93863544 | 10.76866544 | 0.356488217 |
| Egypt | 40.97292949 | 44.80730635 | 0.093583176 | 1090.204957 | 1191.941347 | 0.093318591 | 3.820687817 | 4.517564062 | 0.182395495 |
| El Salvador | 37.12301595 | 38.64986088 | 0.041129334 | 993.6185147 | 1032.564427 | 0.039196041 | 4.121562765 | 4.424155838 | 0.073417073 |
| Equatorial Guinea | 28.23110968 | 28.95029749 | 0.02547501 | 752.294221 | 772.6937322 | 0.0271164 | 3.887100355 | 3.439535297 | −0.115141112 |
| Eritrea | 34.79426389 | 37.80440482 | 0.086512563 | 922.9152361 | 1003.114589 | 0.086897853 | 7.527806927 | 7.003313597 | −0.069674121 |
| Estonia | 166.3377824 | 179.0245537 | 0.076271134 | 4444.354458 | 4777.214206 | 0.07489496 | 18.87312199 | 17.48595169 | −0.073499779 |
| Eswatini | 32.12265964 | 34.36660135 | 0.069855415 | 852.4508544 | 911.6440632 | 0.069438852 | 4.090345484 | 5.394077498 | 0.31873396 |
| Ethiopia | 40.691072 | 34.61907297 | −0.149221899 | 1078.757552 | 919.9697448 | −0.147195083 | 11.8536042 | 5.833924236 | −0.507835411 |
| Fiji | 41.11072659 | 48.74041474 | 0.185588745 | 1102.603206 | 1304.390703 | 0.183010077 | 3.136091502 | 3.797533718 | 0.21091292 |
| Finland | 60.30553217 | 60.71932004 | 0.006861524 | 1605.467203 | 1615.42811 | 0.006204367 | 5.482728997 | 5.225509993 | −0.046914411 |
| France | 65.37502682 | 68.54395801 | 0.048473115 | 1744.132285 | 1823.56647 | 0.04554367 | 5.751487074 | 5.757146915 | 0.000984066 |
| Gabon | 28.71793932 | 32.17143477 | 0.120255685 | 765.454691 | 858.5177478 | 0.121578792 | 3.644957557 | 4.143015367 | 0.136642966 |
| Gambia | 27.74648882 | 28.57341614 | 0.029802954 | 737.5805399 | 759.8739641 | 0.030225071 | 3.905496804 | 4.228704762 | 0.082757194 |
| Georgia | 69.58106302 | 67.65541035 | −0.027674953 | 1842.065362 | 1795.603601 | −0.025222646 | 6.138930021 | 6.585861963 | 0.072802905 |
| Germany | 61.8531313 | 63.61952665 | 0.028557897 | 1648.077456 | 1692.47988 | 0.026941952 | 5.59320726 | 5.400422824 | −0.034467601 |
| Ghana | 29.17962381 | 31.45472112 | 0.077968699 | 775.4683 | 836.5366228 | 0.07875025 | 4.767834609 | 5.133316518 | 0.076655744 |
| Greece | 72.20496105 | 94.26561348 | 0.305528209 | 1924.856534 | 2509.741909 | 0.303859205 | 5.475578484 | 7.006750551 | 0.279636585 |
| Greenland | 42.56064696 | 44.55473779 | 0.046852926 | 1131.024194 | 1180.959507 | 0.044150526 | 3.516183688 | 3.5990197 | 0.0235585 |
| Grenada | 42.48469743 | 69.01120543 | 0.624377943 | 1140.056169 | 1861.786208 | 0.633065334 | 5.880780655 | 11.18190243 | 0.901431644 |
| Guam | 43.32526832 | 52.21556708 | 0.205198931 | 1164.192508 | 1396.034305 | 0.199143866 | 3.554517305 | 3.902099791 | 0.097786129 |
| Guatemala | 37.10522585 | 37.05514687 | −0.001349647 | 993.0221557 | 991.1789764 | −0.001856131 | 7.61432867 | 6.522227605 | −0.143427098 |
| Guinea | 29.05276754 | 28.21857476 | −0.028713023 | 770.9161896 | 750.1279396 | −0.026965642 | 4.063311157 | 3.725804451 | −0.083061989 |

*(Continued)*

 

**Table 2.** (Continued)

| location | prevalence rate (100%) | | | incidence rate (100%) | | | DALYs rate (100%) | | |
|---|---|---|---|---|---|---|---|---|---|
| | 1990 | 2021 | Percentage change | 1990 | 2021 | Percentage change | 1990 | 2021 | Percentage change |
| Guinea-Bissau | 28.71244341 | 28.56465287 | −0.005147265 | 762.8160737 | 759.7179352 | −0.004061449 | 6.668595749 | 5.263698699 | −0.210673596 |
| Guyana | 41.30497747 | 58.44582438 | 0.414982599 | 1108.85027 | 1573.018993 | 0.418603607 | 5.329024013 | 13.73160866 | 1.576758638 |
| Haiti | 42.06671646 | 46.01599862 | 0.093881398 | 1128.564507 | 1237.442498 | 0.096474762 | 7.170853954 | 7.341623261 | 0.023814361 |
| Honduras | 38.97340847 | 37.96457636 | −0.025885139 | 1043.793676 | 1017.387285 | −0.025298478 | 12.6884536 | 10.66245307 | −0.15967277 |
| Hungary | 68.38151646 | 60.03760268 | −0.122020017 | 1822.617902 | 1591.924297 | −0.126572665 | 23.99953114 | 7.560749775 | −0.684962605 |
| Iceland | 56.11575839 | 60.55414993 | 0.079093496 | 1498.67539 | 1613.012347 | 0.07629201 | 4.731454021 | 5.332661704 | 0.127066158 |
| India | 57.82803664 | 67.91901722 | 0.174499796 | 1539.016547 | 1802.02618 | 0.17089461 | 9.832784842 | 8.992457817 | −0.085461753 |
| Indonesia | 69.60287715 | 60.26937613 | −0.134096483 | 1881.908525 | 1622.55859 | −0.13781219 | 9.825995339 | 9.180040083 | −0.065739422 |
| Iran (Islamic Republic of) | 41.77135373 | 49.23175151 | 0.178600814 | 1112.176096 | 1306.117711 | 0.174380312 | 6.321792753 | 5.81546534 | −0.080092378 |
| Iraq | 40.09212397 | 45.08832674 | 0.124618062 | 1069.633681 | 1199.818005 | 0.12170926 | 4.203873269 | 4.304992573 | 0.024053842 |
| Ireland | 64.71546673 | 69.39441709 | 0.072300342 | 1727.605543 | 1845.906615 | 0.0684769 | 5.773408174 | 5.729529494 | −0.007600135 |
| Israel | 61.2698637 | 64.21936748 | 0.048139552 | 1636.581783 | 1712.608232 | 0.046454414 | 5.947395734 | 5.91922889 | −0.004735996 |
| Italy | 86.45583189 | 89.58408025 | 0.036183196 | 2310.825303 | 2394.537843 | 0.036226252 | 7.839292859 | 7.299513241 | −0.068855652 |
| Jamaica | 40.86463917 | 48.70758827 | 0.191925079 | 1095.151506 | 1305.542886 | 0.192111666 | 4.012398148 | 8.493503029 | 1.116814612 |
| Japan | 84.5350764 | 87.2566115 | 0.032194152 | 2255.621468 | 2324.408362 | 0.030495761 | 6.758668337 | 7.744766917 | 0.145901312 |
| Jordan | 38.61538587 | 58.4022871 | 0.512409776 | 1030.080515 | 1555.587713 | 0.510161284 | 3.049449937 | 4.502851937 | 0.476611202 |
| Kazakhstan | 91.00180451 | 100.8650484 | 0.108385147 | 2423.637631 | 2697.567953 | 0.113024455 | 16.77504305 | 18.85062419 | 0.123730302 |
| Kenya | 34.49181673 | 38.06757907 | 0.103669875 | 916.7168164 | 1010.836319 | 0.102670204 | 4.557705183 | 5.944788392 | 0.304338072 |
| Kiribati | 40.42694253 | 44.53550791 | 0.101629387 | 1088.007618 | 1196.355614 | 0.099583857 | 3.742299863 | 3.872844504 | 0.034883533 |
| Kuwait | 45.52867359 | 51.34758172 | 0.127807548 | 1212.97611 | 1362.432609 | 0.12321471 | 3.413275181 | 4.192073553 | 0.228167473 |
| Kyrgyzstan | 86.033492 | 83.2129914 | −0.032783751 | 2284.598919 | 2207.341816 | −0.033816484 | 9.912846426 | 8.801405614 | −0.112121258 |
| Lao People's Democratic Republic | 56.87503073 | 58.29351507 | 0.024940371 | 1531.598943 | 1566.834223 | 0.023005552 | 10.6084072 | 7.834819843 | −0.261451819 |
| Latvia | 170.7725605 | 176.5272215 | 0.033697808 | 4583.385531 | 4714.473381 | 0.02860066 | 23.5486248 | 22.64649323 | −0.03830931 |
| Lebanon | 40.92933865 | 45.6148592 | 0.114478286 | 1087.416238 | 1213.173668 | 0.115647923 | 4.946631187 | 4.404328973 | −0.109630614 |
| Lesotho | 32.79592492 | 33.94037745 | 0.034896181 | 868.9315156 | 900.4741681 | 0.036300505 | 3.3230576 | 5.295254842 | 0.593488732 |
| Liberia | 28.34851065 | 30.23723284 | 0.066625094 | 752.997062 | 803.3578581 | 0.066880468 | 4.147623789 | 4.286130959 | 0.033394343 |
| Libya | 41.3972483 | 48.39049821 | 0.168930308 | 1102.995159 | 1284.52671 | 0.164580551 | 3.482678583 | 5.933420263 | 0.703694476 |
| Lithuania | 173.8140997 | 175.0230425 | 0.006955378 | 4669.55289 | 4670.876054 | 0.00028336 | 21.42325171 | 21.71671704 | 0.01369845 |
| Luxembourg | 74.6456317 | 77.28463588 | 0.035353766 | 1991.769663 | 2057.772615 | 0.033137844 | 6.142708431 | 6.096559801 | −0.00751275 |
| Madagascar | 34.06208668 | 36.61471721 | 0.074940521 | 904.0021265 | 971.7973569 | 0.074994547 | 5.257786195 | 5.03829156 | −0.041746588 |
| Malawi | 33.48231501 | 34.40910659 | 0.027680033 | 889.5729619 | 914.934033 | 0.028509265 | 5.226792486 | 5.589169674 | 0.069330701 |
| Malaysia | 55.89129651 | 61.63496418 | 0.102764975 | 1502.513132 | 1654.258696 | 0.100994501 | 5.064877422 | 5.68674214 | 0.122779816 |
| Maldives | 53.18856251 | 72.76060472 | 0.367974641 | 1430.185869 | 1956.046672 | 0.367687036 | 4.169537104 | 5.519324714 | 0.32372601 |
| Mali | 29.64489597 | 28.4613595 | −0.039923785 | 786.6189495 | 756.9204839 | −0.037754577 | 5.009572441 | 3.820801297 | −0.237299921 |
| Malta | 72.55625782 | 74.8416269 | 0.031497891 | 1936.106588 | 1993.345161 | 0.029563751 | 6.664849537 | 6.587488649 | −0.011607297 |
| Marshall Islands | 39.69917204 | 49.08462602 | 0.236414351 | 1067.723206 | 1316.01509 | 0.232543305 | 3.400282983 | 4.114226312 | 0.209965856 |
| Mauritania | 29.01012305 | 29.58731672 | 0.019896285 | 770.6023479 | 786.1058039 | 0.02011862 | 5.360874408 | 4.046984945 | −0.245088648 |
| Mauritius | 55.97965345 | 70.27193902 | 0.255312148 | 1505.990237 | 1881.424132 | 0.249293711 | 4.539134732 | 5.582644923 | 0.229891874 |
| Mexico | 52.35460062 | 73.47258352 | 0.403364416 | 1409.803287 | 1969.136961 | 0.3967459 | 10.49242554 | 12.32720581 | 0.174867123 |

*(Continued)*

**Table 2.** (Continued)

| location | prevalence rate (100%) | | | incidence rate (100%) | | | DALYs rate (100%) | | |
|---|---|---|---|---|---|---|---|---|---|
| | 1990 | 2021 | Percentage change | 1990 | 2021 | Percentage change | 1990 | 2021 | Percentage change |
| Micronesia (Federated States of) | 41.95234833 | 49.35889744 | 0.176546711 | 1128.088658 | 1325.121825 | 0.174661066 | 3.82375065 | 4.111375214 | 0.075220533 |
| Monaco | 70.43113243 | 71.90632666 | 0.020945201 | 1873.53035 | 1908.927862 | 0.018893482 | 5.35430359 | 5.538309693 | 0.03436602 |
| Mongolia | 69.83752654 | 75.69350833 | 0.083851506 | 1854.806321 | 2005.323415 | 0.081149764 | 7.128499139 | 6.882687466 | −0.034482949 |
| Montenegro | 54.24864477 | 54.47320303 | 0.004139426 | 1433.495075 | 1438.485455 | 0.003481268 | 4.424747741 | 4.262779659 | −0.036605043 |
| Morocco | 39.07355021 | 45.47177153 | 0.163748144 | 1040.674695 | 1208.023634 | 0.160808118 | 3.316175482 | 4.866782971 | 0.467589094 |
| Mozambique | 36.13459623 | 35.93722983 | −0.005461979 | 958.2859037 | 954.2086874 | −0.004254697 | 6.093733316 | 7.409806519 | 0.215971578 |
| Myanmar | 57.29599617 | 65.48814305 | 0.142979395 | 1543.859119 | 1759.203739 | 0.139484631 | 10.37387599 | 8.918912178 | −0.14025267 |
| Namibia | 32.7804881 | 34.50051126 | 0.052470944 | 869.760788 | 914.1596363 | 0.051047195 | 3.516125575 | 3.792652942 | 0.078645475 |
| Nauru | 46.14817527 | 48.43311431 | 0.049513096 | 1240.103402 | 1300.964264 | 0.049077248 | 4.639668015 | 4.364472858 | −0.059313545 |
| Nepal | 50.71865831 | 52.68677705 | 0.038804629 | 1347.486789 | 1400.528238 | 0.039363242 | 7.616834286 | 6.566067241 | −0.13795325 |
| Netherlands | 67.04146729 | 69.75626426 | 0.040494295 | 1790.937538 | 1857.49443 | 0.037163157 | 7.078431589 | 6.811865796 | −0.037658878 |
| New Zealand | 71.23558735 | 67.27456877 | −0.055604491 | 1914.779195 | 1794.951256 | −0.062580552 | 8.214009221 | 7.229124041 | −0.119903101 |
| Nicaragua | 35.63527826 | 39.03684689 | 0.095455088 | 955.2288108 | 1043.754056 | 0.092674388 | 4.155244809 | 4.524898488 | 0.088960746 |
| Niger | 28.69594455 | 27.72858432 | −0.033710695 | 762.073199 | 737.7043945 | −0.031976987 | 3.918483401 | 3.137823962 | −0.199224894 |
| Nigeria | 31.41262214 | 29.34183993 | −0.065921979 | 833.4667105 | 779.1226599 | −0.065202425 | 3.934489016 | 3.453983105 | −0.122126637 |
| Niue | 47.58997576 | 55.71113067 | 0.170648435 | 1274.955664 | 1489.49913 | 0.168275236 | 3.914190989 | 4.370699209 | 0.11662901 |
| North Macedonia | 53.98176387 | 54.64261584 | 0.012242134 | 1425.847618 | 1442.703399 | 0.011821586 | 4.360897752 | 4.165656147 | −0.044770966 |
| Northern Mariana Islands | 44.76921822 | 58.09642284 | 0.297686784 | 1200.996753 | 1550.312202 | 0.290854615 | 3.623947021 | 4.353870309 | 0.201416655 |
| Norway | 112.6750703 | 117.8814135 | 0.0462067 | 3017.018762 | 3147.297466 | 0.043181271 | 9.102739165 | 9.101252533 | −0.000163317 |
| Oman | 47.57143915 | 53.76618969 | 0.130219952 | 1268.155347 | 1431.45608 | 0.128770291 | 3.685086667 | 4.33996586 | 0.177710662 |
| Pakistan | 54.43413042 | 53.65220714 | −0.014364577 | 1447.778637 | 1426.557229 | −0.014657909 | 8.825411645 | 9.599796069 | 0.08774485 |
| Palau | 43.13529015 | 64.35867145 | 0.492018976 | 1158.523938 | 1717.961357 | 0.482888096 | 3.723534144 | 5.109335669 | 0.372173712 |
| Palestine | 36.71087114 | 41.97246274 | 0.143325163 | 980.0907976 | 1118.459551 | 0.141179525 | 3.089561514 | 3.380411182 | 0.094139465 |
| Panama | 38.5588015 | 41.75553305 | 0.082905366 | 1031.488012 | 1113.445354 | 0.079455448 | 4.195260715 | 4.457897838 | 0.06260329 |
| Papua New Guinea | 40.36109036 | 43.55040031 | 0.07901942 | 1083.334801 | 1167.49669 | 0.077687792 | 3.245628802 | 3.366628041 | 0.037280677 |
| Paraguay | 40.00645756 | 41.21992586 | 0.030331811 | 1065.531771 | 1097.305142 | 0.029819262 | 3.833731664 | 5.673185367 | 0.47980763 |
| Peru | 85.67717131 | 92.51102381 | 0.079762817 | 2288.948532 | 2463.073886 | 0.076072202 | 7.332507715 | 7.829213602 | 0.067740247 |
| Philippines | 92.10947492 | 108.3345759 | 0.176150184 | 2488.255124 | 2913.405103 | 0.170862696 | 17.73309124 | 20.69901675 | 0.167253722 |
| Poland | 69.41088809 | 39.04640589 | −0.437459929 | 1853.619863 | 1029.132813 | −0.444798347 | 11.84908443 | 3.877717965 | −0.672741131 |
| Portugal | 57.20723997 | 69.9143391 | 0.222123968 | 1524.081318 | 1863.378998 | 0.222624394 | 5.483153231 | 5.853469716 | 0.06753714 |
| Puerto Rico | 46.58592485 | 54.92377763 | 0.178977938 | 1245.530896 | 1467.968915 | 0.178588921 | 4.144850269 | 7.571277177 | 0.826670853 |
| Qatar | 51.87525818 | 58.62053448 | 0.130028775 | 1381.996704 | 1559.986181 | 0.128791535 | 4.340515095 | 4.627783802 | 0.066183091 |
| Republic of Korea | 63.20357947 | 80.12148644 | 0.267673241 | 1693.177842 | 2132.996539 | 0.259759304 | 5.254435029 | 6.289290758 | 0.196949001 |
| Republic of Moldova | 151.5526666 | 168.1513024 | 0.109523878 | 4052.515796 | 4488.62593 | 0.107614666 | 16.43211114 | 18.28526954 | 0.112776647 |
| Romania | 55.8716897 | 57.70414168 | 0.032797504 | 1476.459032 | 1523.675795 | 0.031979731 | 4.252495434 | 4.316759275 | 0.01511203 |
| Russian Federation | 200.7863189 | 178.2320393 | −0.112329763 | 5399.890317 | 4765.511931 | −0.117479865 | 26.70671615 | 21.87316307 | −0.180986425 |
| Rwanda | 34.09089352 | 35.98255344 | 0.055488717 | 904.6261113 | 955.9400571 | 0.056723927 | 5.998578097 | 4.624446643 | −0.229076196 |

*(Continued)*

| location | prevalence rate (100%) | | | incidence rate (100%) | | | DALYs rate (100%) | | |
|---|---|---|---|---|---|---|---|---|---|
| | 1990 | 2021 | Percentage change | 1990 | 2021 | Percentage change | 1990 | 2021 | Percentage change |
| Saint Kitts and Nevis | 39.51791737 | 56.42757769 | 0.42789857 | 1059.950211 | 1511.528285 | 0.426037062 | 3.714274386 | 5.305879273 | 0.428510315 |
| Saint Lucia | 40.71572575 | 60.25138924 | 0.479806343 | 1091.616144 | 1612.123257 | 0.476822476 | 5.006991724 | 10.18660809 | 1.034476718 |
| Saint Vincent and the Grenadines | 41.04617234 | 64.18790759 | 0.563797644 | 1101.285134 | 1724.519986 | 0.565915977 | 5.07007967 | 8.629813256 | 0.702106045 |
| Samoa | 40.4939468 | 47.87763796 | 0.182340615 | 1088.687316 | 1283.254686 | 0.178717403 | 3.356555333 | 3.866654958 | 0.151971165 |
| San Marino | 65.3680718 | 73.89973657 | 0.130517308 | 1743.375958 | 1961.832502 | 0.125306618 | 5.51470884 | 6.098336316 | 0.105831059 |
| Sao Tome and Principe | 28.21922878 | 31.36123237 | 0.111342646 | 749.5648446 | 832.4218333 | 0.110540121 | 4.163067539 | 4.569327522 | 0.09758669 |
| Saudi Arabia | 44.19001058 | 52.14820074 | 0.180090252 | 1177.488146 | 1385.883701 | 0.176983145 | 3.33110903 | 4.009792845 | 0.203741099 |
| Senegal | 28.11210199 | 29.04215331 | 0.033083663 | 746.872523 | 771.756751 | 0.033317905 | 4.311159557 | 3.943574588 | −0.085263596 |
| Serbia | 54.95864012 | 54.09931662 | −0.015635822 | 1451.091818 | 1428.158975 | −0.015803854 | 5.056446396 | 4.561165399 | −0.09795041 |
| Seychelles | 54.49594806 | 78.02592936 | 0.431774877 | 1466.426782 | 2091.8068 | 0.426465218 | 5.442145411 | 8.635776391 | 0.586833085 |
| Sierra Leone | 28.05291967 | 28.51851025 | 0.016596867 | 745.1880954 | 758.2635067 | 0.017546457 | 3.54984224 | 3.611612163 | 0.017400752 |
| Singapore | 64.05672727 | 80.10034541 | 0.250459535 | 1719.425307 | 2136.139184 | 0.242356487 | 5.372209908 | 6.595134626 | 0.227639042 |
| Slovakia | 62.12510188 | 62.07741432 | −0.000767605 | 1642.898803 | 1639.59198 | −0.002012797 | 7.449935856 | 5.418033224 | −0.272740957 |
| Slovenia | 52.47415612 | 54.03031977 | 0.029655811 | 1385.724625 | 1426.087739 | 0.029127803 | 6.760963821 | 4.434124219 | −0.344157973 |
| Solomon Islands | 42.98779135 | 50.2415615 | 0.168740238 | 1155.242433 | 1347.312887 | 0.166259868 | 3.617096679 | 4.116995109 | 0.138204332 |
| Somalia | 36.66949058 | 37.45281319 | 0.021361699 | 973.6868593 | 997.0309911 | 0.023974989 | 8.443973277 | 7.832707609 | −0.072390763 |
| South Africa | 36.70346051 | 37.16780683 | 0.012651295 | 973.1120915 | 983.4996605 | 0.010674586 | 4.433506471 | 4.119844056 | −0.070748158 |
| South Sudan | 32.43713766 | 36.20601928 | 0.116190327 | 862.3845806 | 961.4198161 | 0.114838829 | 6.153759854 | 7.393245243 | 0.2014192 |
| Spain | 57.0745409 | 88.32575976 | 0.547550946 | 1522.024125 | 2342.821095 | 0.539279869 | 5.310948872 | 7.19780194 | 0.355276075 |
| Sri Lanka | 58.1867462 | 67.20022609 | 0.154906065 | 1563.98441 | 1799.430296 | 0.150542348 | 4.923295159 | 5.596434595 | 0.136725387 |
| Sudan | 38.5256263 | 40.33694665 | 0.047015987 | 1026.188137 | 1075.17885 | 0.047740479 | 3.385648575 | 4.399255017 | 0.299383241 |
| Suriname | 43.55324361 | 69.92980073 | 0.605616366 | 1168.733049 | 1893.299787 | 0.619959142 | 5.44395068 | 10.72541032 | 0.970152 |
| Sweden | 71.10415018 | 62.92451688 | −0.115037354 | 1897.707509 | 1679.589292 | −0.114937742 | 6.344295322 | 5.185795725 | −0.182604929 |
| Switzerland | 72.25691966 | 74.94012508 | 0.037134235 | 1927.44251 | 1994.322921 | 0.034699043 | 6.069901571 | 5.979668938 | −0.014865584 |
| Syrian Arab Republic | 38.82983776 | 48.21881018 | 0.241797879 | 1036.050864 | 1278.028329 | 0.233557514 | 4.343664194 | 4.758406377 | 0.095482101 |
| Taiwan (Province of China) | 47.63197442 | 74.57167198 | 0.565580115 | 1281.950973 | 2001.865818 | 0.561577517 | 5.232390696 | 7.569459314 | 0.446654073 |
| Tajikistan | 72.55581479 | 78.43412861 | 0.081017818 | 1926.49601 | 2078.83476 | 0.079075559 | 11.20995694 | 9.484532977 | −0.153918875 |
| Thailand | 70.88364859 | 82.2992149 | 0.161046539 | 1916.451304 | 2207.261843 | 0.151744288 | 16.27176281 | 15.32378788 | −0.058258896 |
| Timor-Leste | 52.45229418 | 53.42952388 | 0.018630828 | 1411.013363 | 1436.272742 | 0.017901588 | 6.711675657 | 6.288613372 | −0.06303378 |
| Togo | 27.60034838 | 29.90382883 | 0.083458383 | 733.735946 | 794.1440717 | 0.082329516 | 4.09557063 | 4.373639952 | 0.067895135 |
| Tokelau | 42.44885973 | 50.95939823 | 0.200489214 | 1139.006513 | 1363.385953 | 0.196995748 | 3.625742356 | 4.079412671 | 0.125124808 |
| Tonga | 40.52213223 | 47.10650634 | 0.162488343 | 1088.011558 | 1263.143929 | 0.160965543 | 3.288375672 | 3.72328836 | 0.132257604 |
| Trinidad and Tobago | 48.93189438 | 94.46597909 | 0.930560431 | 1317.900677 | 2580.309945 | 0.957894089 | 10.92401919 | 27.54553346 | 1.521556671 |
| Tunisia | 39.45408432 | 47.98636608 | 0.216258517 | 1050.94726 | 1273.164731 | 0.211444931 | 3.176285432 | 4.372110647 | 0.376485439 |
| Türkiye | 40.78850178 | 48.67618345 | 0.193380029 | 1085.615177 | 1293.216814 | 0.191229491 | 4.376245686 | 4.3999352 | 0.005413205 |
| Turkmenistan | 69.17569546 | 77.08700695 | 0.114365478 | 1836.01438 | 2044.3766 | 0.113486159 | 6.39579441 | 7.58541186 | 0.185999951 |

*(Continued)*

**Table 2.** (Continued)

| location | prevalence rate (100%) | | | incidence rate (100%) | | | DALYs rate (100%) | | |
|---|---|---|---|---|---|---|---|---|---|
| | 1990 | 2021 | Percentage change | 1990 | 2021 | Percentage change | 1990 | 2021 | Percentage change |
| Tuvalu | 43.42413766 | 48.81071612 | 0.124045721 | 1165.662409 | 1308.73577 | 0.122739962 | 3.896839709 | 3.941534542 | 0.011469508 |
| Uganda | 32.09652675 | 33.55142132 | 0.045328723 | 852.5142524 | 892.2043872 | 0.046556565 | 4.32498951 | 4.771367579 | 0.103209052 |
| Ukraine | 202.3377859 | 204.1609286 | 0.009010392 | 5456.396241 | 5474.29369 | 0.003280086 | 21.04431209 | 19.00975891 | −0.096679482 |
| United Arab Emirates | 50.96047727 | 69.80710655 | 0.369828351 | 1357.501382 | 1847.501385 | 0.360957278 | 4.086004492 | 5.359776021 | 0.31174012 |
| United Kingdom | 77.19285178 | 91.77862993 | 0.188952446 | 2061.707967 | 2445.73879 | 0.186268293 | 7.161949432 | 8.897317611 | 0.242303886 |
| United Republic of Tanzania | 33.07579998 | 35.90521829 | 0.085543458 | 878.1338881 | 953.3016257 | 0.085599404 | 5.717559876 | 5.371588493 | −0.060510321 |
| United States of America | 89.34459266 | 44.67776435 | −0.499938798 | 2382.968573 | 1184.240857 | −0.503039667 | 7.542666497 | 5.284633712 | −0.299367974 |
| United States Virgin Islands | 57.72600474 | 103.912036 | 0.800090557 | 1548.892784 | 2826.640355 | 0.824942555 | 10.92428542 | 12.92862584 | 0.183475655 |
| Uruguay | 100.6663733 | 102.4224638 | 0.017444659 | 2675.456054 | 2718.811926 | 0.01620504 | 8.338424497 | 9.813387491 | 0.176887492 |
| Uzbekistan | 70.45913901 | 75.80882038 | 0.075926011 | 1870.644148 | 2009.762992 | 0.074369486 | 5.243773677 | 5.634312396 | 0.074476654 |
| Vanuatu | 40.39242513 | 44.72044647 | 0.107149331 | 1084.425019 | 1199.191914 | 0.105832024 | 3.30929819 | 3.566839007 | 0.077823394 |
| Venezuela (Bolivarian Republic of) | 39.0150887 | 55.64976736 | 0.426365266 | 1045.48954 | 1485.361609 | 0.420733113 | 6.246315163 | 11.12454896 | 0.780977852 |
| Viet Nam | 54.23409094 | 92.46298294 | 0.704886748 | 1461.943914 | 2479.014361 | 0.695697309 | 5.216992761 | 8.10388306 | 0.553362911 |
| Yemen | 39.18689031 | 41.02915745 | 0.047012333 | 1043.756671 | 1093.367023 | 0.047530572 | 3.255259947 | 4.098195997 | 0.258945849 |
| Zambia | 35.01036908 | 37.6237028 | 0.074644564 | 929.1776504 | 999.9039587 | 0.0761171 | 6.671448706 | 7.276963541 | 0.090762121 |
| Zimbabwe | 33.36746206 | 35.28587289 | 0.05749346 | 884.0001448 | 931.4965104 | 0.053728912 | 3.217794929 | 4.436098984 | 0.378614574 |

APC = −59.52%; 95% CI: −79.58% to −39.42%), while a notable increase in incidence was recorded from 1995 to 2003 (APC = 21.51%; 95% CI: 18.61% to 24.40%) (Fig 7B and Supplementary S11 Table). Changes in male incidence significantly contributed to these trends.Disability-adjusted life years (DALYs) serve as a crucial metric for assessing health burden, providing substantial support for public health policy formulation and medical research by quantifying health loss attributable to disease or injury.From 1990 to 2021, there was an overall significant decrease in DALYs among individuals aged 20–54 years due to urolithiasis (AAPC = −53.06%; 95% CI: −62.06% to −44.05%; P < 0.001). The most significant declines were noted between 1990 and 1994 and from 2008 to 2012 (APC = −84.50%; 95% CI: −106.43% to −62.51%, P < 0.001 and APC = −87.48%; 95% CI: −121.66% to −53.18%). Importantly, DALYs in females continued to rise between 2012 and 2021 (APC = 72.13%; 95% CI: 26.39% to 118.07%, and APC = 21.58%; 95% CI: 1.16% to 42.05%), while a gradual decline was observed among males (Fig 7C and Supplementary S12 Table). The global prevalence, incidence, and DALYs of urolithiasis among individuals aged 20–54 declined dramatically between 1990 and 2021, but there were sporadic variations. The years 1995–2003 had a modest spike in both prevalence and incidence, while 1990–1994 and 2008–2012 saw the largest declines in DALYs. However, from 2012 to 2021, female DALYs increased more, suggesting that the gender gap was getting worse.

## The association between urolithiasis burden and SDI

In 2021, there was a positive correlation between the prevalence, incidence, and DALY rates of urolithiasis in the 20–54 age group and the Sociodemographic Index (SDI). The overall burden of disease is increasing as the economy improves.

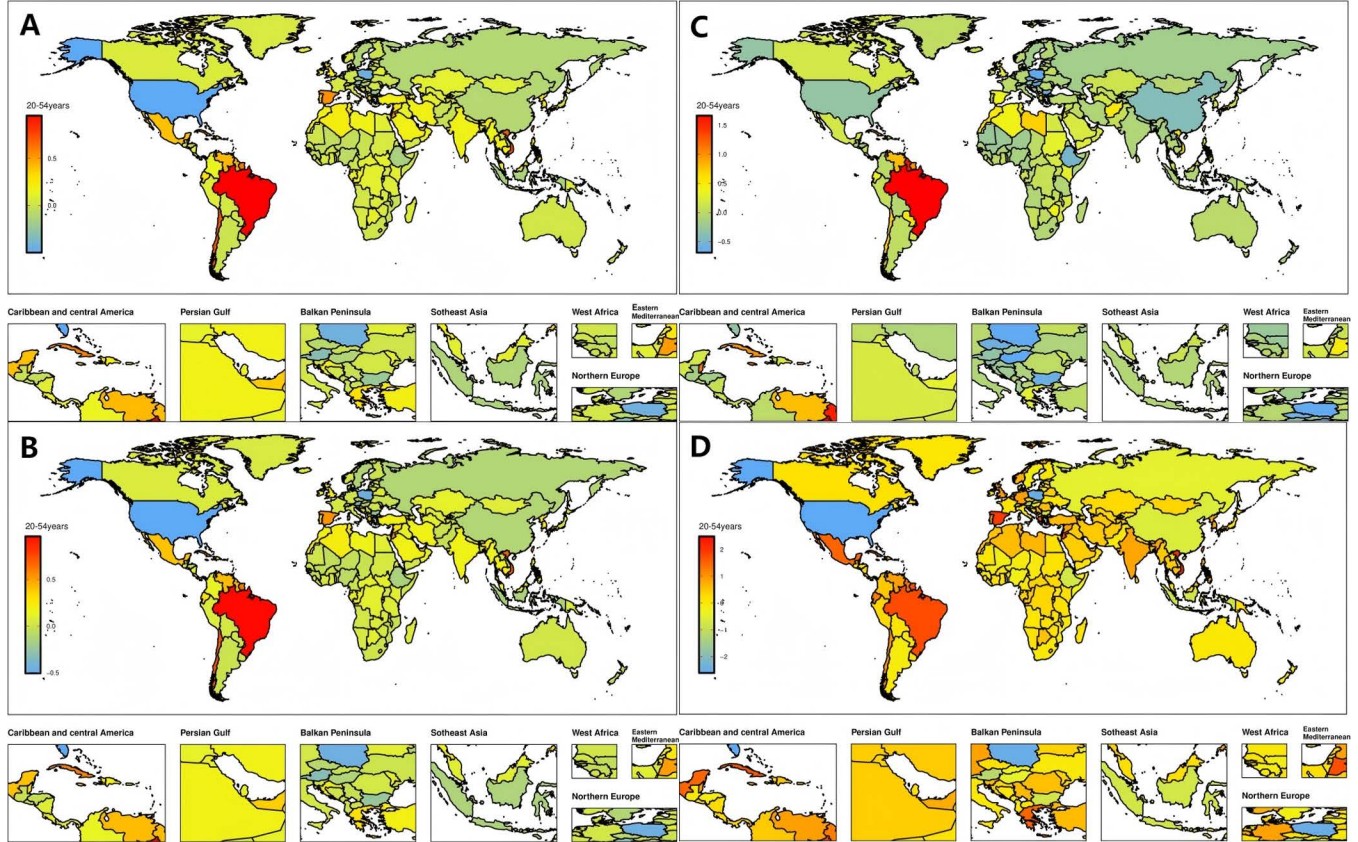

**Fig 4. Temporal trend of migraine burden in 20-54years globally.** (A) Percentage change in prevalent rate across 204 countries in 1990 and 2021. (B) Percentage change in incidence rate across 204 countries in 1990 and 2021. (C) Percentage change in DALYs rate cases across 204 countries in 1990 and 2021. (D) EAPC in prevalent rates across 204 countries from 1990 to 2021. EAPC = Estimated Annual Percentage Change.

In 21 regions, the most significant rise in urolithiasis burden was observed in the 20–54 age group when SDI values ranged from 0.6 to 0.7. The burden of urolithiasis peaked when the SDI reached 0.7. However, when the SDI value exceeded 0.7, the prevalence and incidence of urolithiasis were negatively correlated with the DALY rates in the 20–54 age group. Notably, Eastern European countries exhibited significantly higher prevalence and DALY rates compared to other regions (Fig 8A–C). The greatest rise in illness burden was found in 21 regions with SDIs between 0.6 and 0.7, which are primarily moderate. Regional disparities in prevention and control as well as unequal access to medical resources may be the reason why the prevalence and DALY rate in Eastern European nations were substantially higher than those in other regions.

## Discussion

As part of the new Agenda 2030, the United Nations set the Sustainable Development Goals (SDGs) in 2015 with the aim of "promoting physical and mental health and well-being, extending life expectancy for all individuals, and ensuring universal health coverage and access to quality healthcare." By 2030, the ultimate objective is to lower mortality and reduce the burden of disease on the world's population [20]. The worldwide society is becoming aware that urinary tract disorders pose a serious danger to global health. Urolithiasis, in particular, is becoming increasingly common, posing significant difficulties to global public health policies and healthcare systems. A recent assessment of epidemiological data from seven

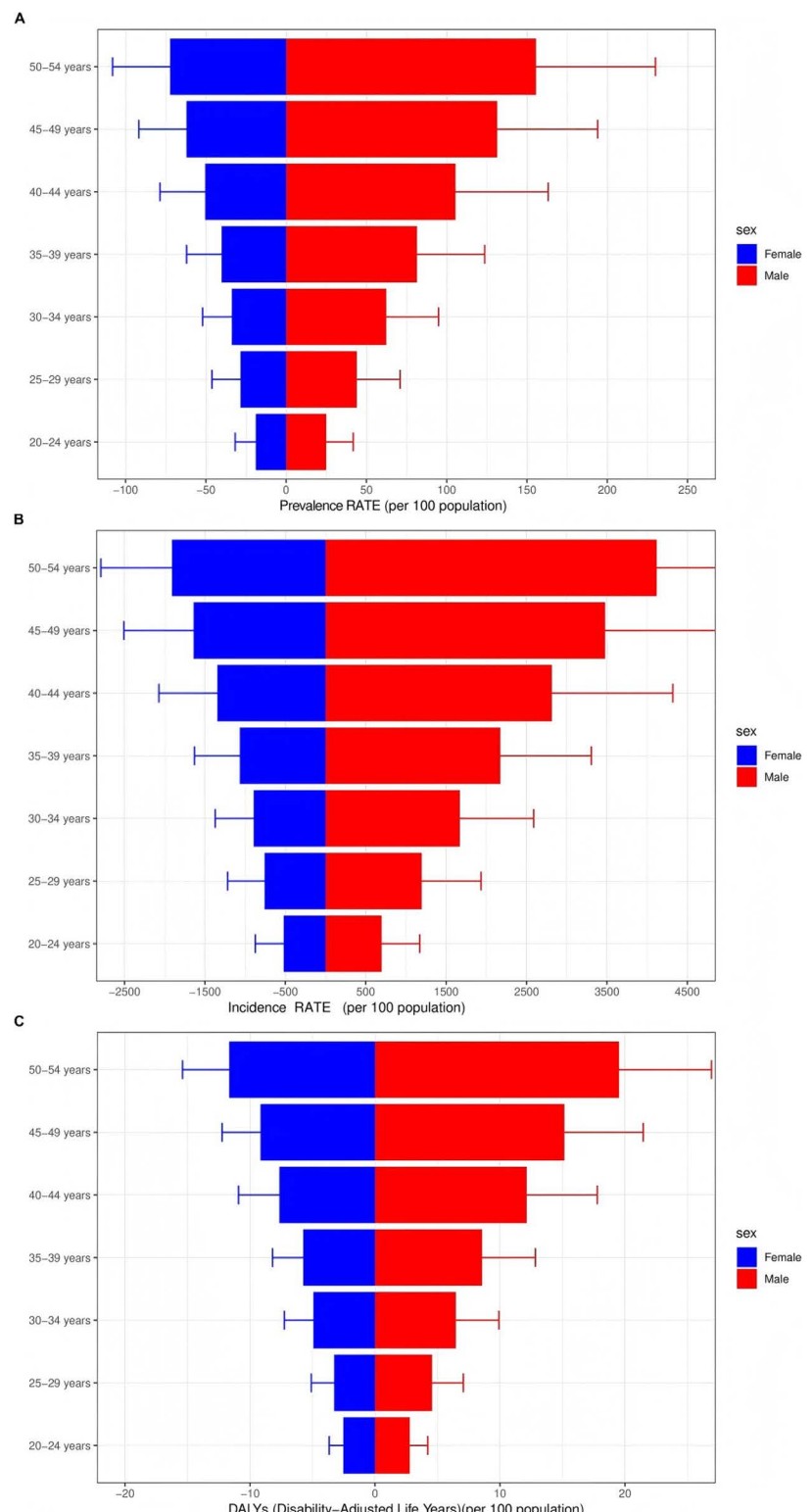

**Fig 5. Sex- and age-structured analysis of urolithiasis disease burden in 2021.** (A) Prevalence rates; **(B)** Incidence rates; **(C)** DALYs rates.

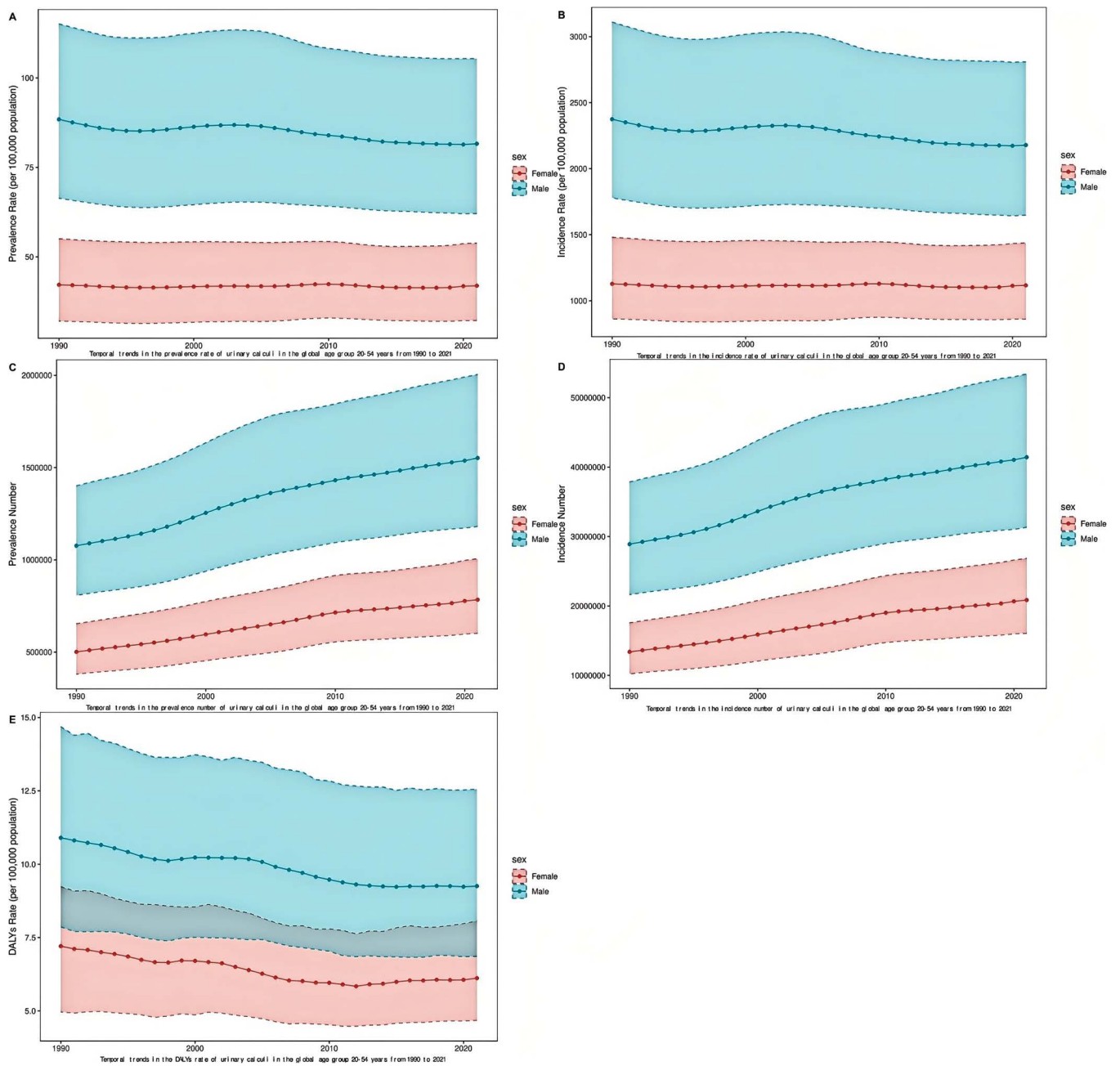

**Fig 6. Global temporal trends in urolithiasis disease burden, 1990–2021.** (A) Prevalence rates in 20-54 age groups. (B) Incidence rates in 20-54 age groups. (C) Prevalence number; (D) Incidence number; (E) DALYs rate.

nations found that kidney stone incidence rates ranged from 114 to 720 per 100,000 people, with prevalence rates ranging from 1.7% to 14.8%. Notably, these rates appear to be increasing in almost all investigated countries [21]. According to statistics from the National Health and Nutrition Examination Survey (NHANES), the prevalence of self-reported kidney stones in the United States than quadrupled, rising from 3.2% in 1976–1980 to 8.8% in 2007–2010 [22,23]. Furthermore, from 2000 to 2010, the lifetime prevalence of kidney stones in the United Kingdom increased by 63%, from 7.14% to

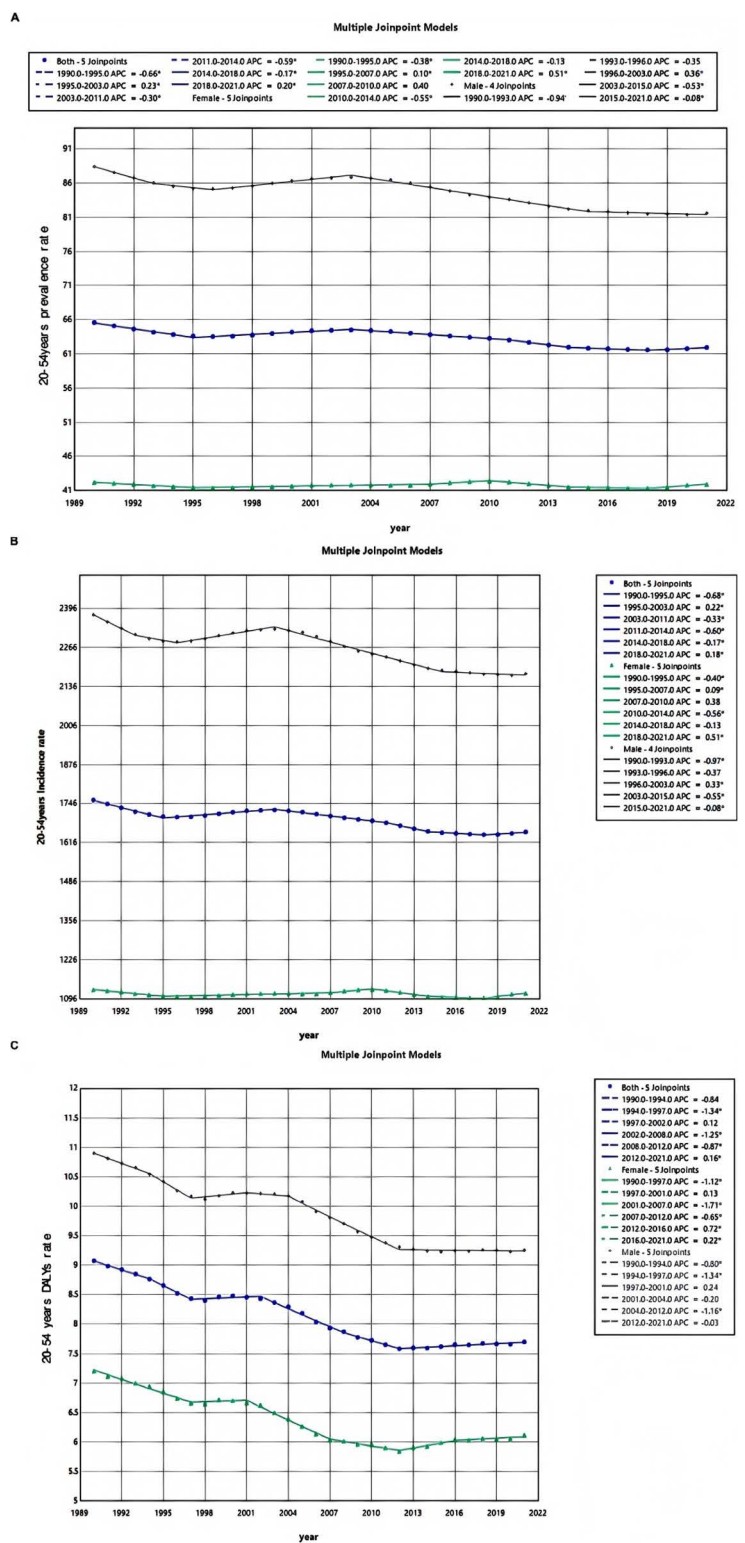

**Fig 7. Joinpoint regression analysis of the urolithiasis disease burden temporal trends, 1990–2021.** (A) 20-54years prevalence rates; (B) 20-54years incidence rates; (C) 20-54years DALYs rates.

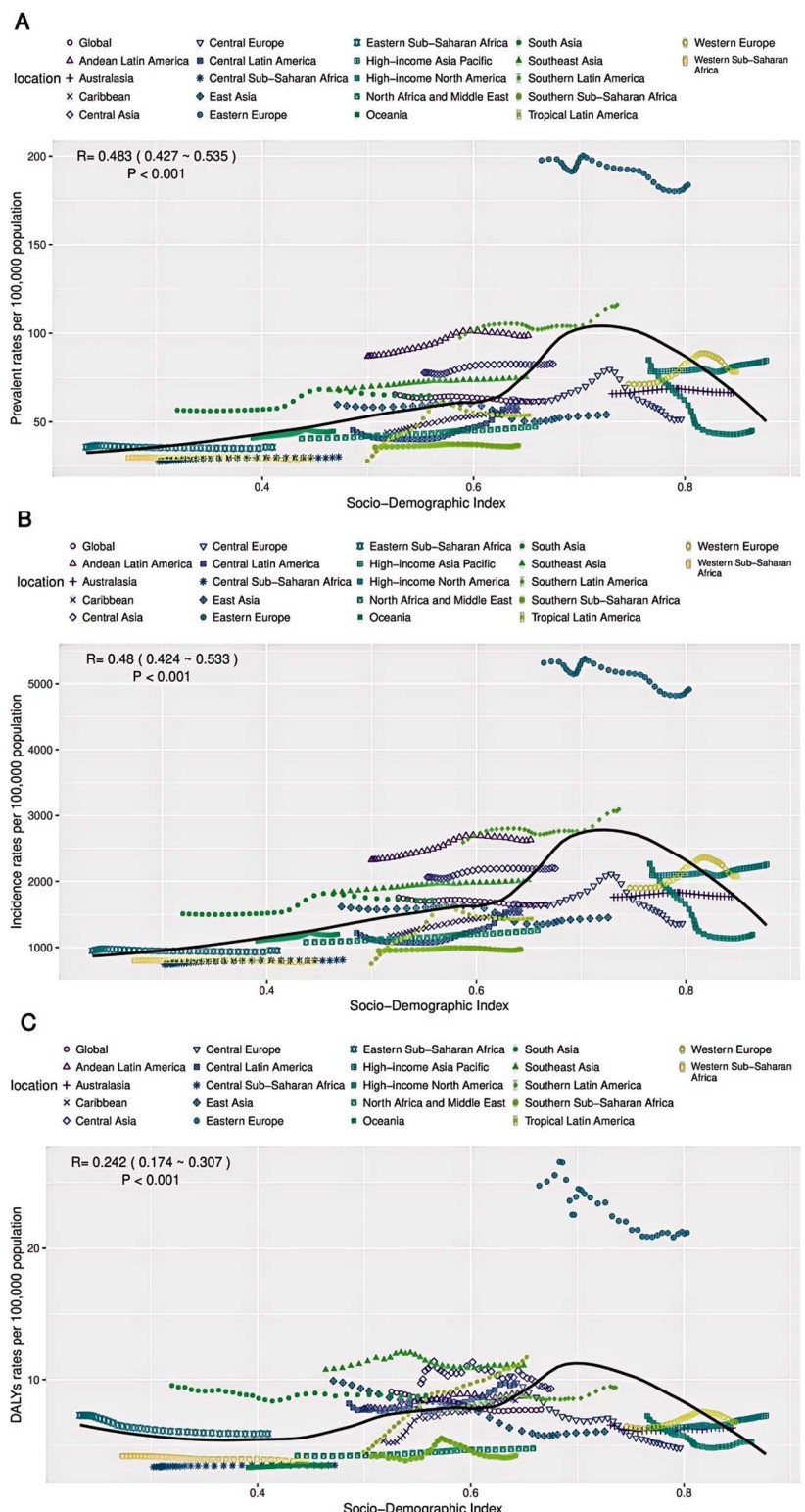

**Fig 8. The associations between the SDI and prevalent, incidence, DALYs rates per 100,000 population of migraine in 20-54years across 21 GBD regions.** SDI = Socio-Demographic Index, GBD = Global Burden of Disease (A) prevalent (B) inceidence **(C)** DALYs.

11.62% [24]. The aim of this study is to conduct a comprehensive analysis of global trends and patterns in the prevalence, incidence, and disability-adjusted life years (DALYs) associated with urolithiasis for individuals aged 20–54. Our findings indicate that the overall worldwide burden, measured in terms of both the total number of cases and the population afflicted, has grown since 1990, which is consistent with current literature across all age categories [25].

However, it is critical to recognize that these trends range across countries, gender classifications, and geographical regions.

This study seeks to provide a thorough assessment of the estimated burden and trends related to urolithiasis among people aged 20–54 at the national, regional, and worldwide levels from 1990 to 2021 using data from the Global Burden of Disease (GBD) 2021. According to a number of studies, working-age persons have a significantly higher frequency of stone disease, while older populations have a lower incidence [13,26,27]. According to a French study, the highest prevalence of urolithiasis was seen in women aged 30–39, whereas the highest incidence occurred in men aged 40–49. According to calculations, the overall male-to-female (M/F) ratio was 2.28 [27]. Corresponding data from an epidemiological examination in Germany that used telephone interviews, the prevalence of urolithiasis was 9.7% in males and 5.9% in women over the age of 5 [28]. A cohort study in Korea showed that the highest incidence of urolithiasis occurred at ages 50–54 years for men and 55–59 years for women [29]. Multiple studies have found that the occurrence of stones has increased significantly among both men and women aged 25 and up, with men showing a particularly noticeable tendency. Furthermore, a study of the age distribution of stone patients between 1979 and 2001 demonstrated that the observed rise in prevalence and incidence was primarily due to the occurrence of stones in the older age group (>50 years). Conversely, there is also evidence indicating that an increasing number of young women are now affected by urinary calculi [28]. Undoubtedly, a comprehensive understanding of urinary stone prevalence trends in the 20–54 age group is essential to assess the potential to achieve related health goals. However, a comprehensive analysis of the prevalence, incidence and DALYs of urinary stones in this population in different countries and regions of the world is lacking. Therefore, we believe that there is a need for timely strengthening and updating of the global data on urinary stone burden in the 20–54 age group so that policy makers can be informed and develop effective prevention and control strategies. The study is the first to provide a comprehensive estimate of the prevalence, incidence and DALYs of urinary stones in patients aged 20–54 years over the past 32 years using GBD 2021 data on a global scale. Over the past 32 years, there has been a notable rise in the incidence of urinary stones, the number of reported cases, and the associated disability-adjusted life years (DALYs) within the global 20–54 age group, with percentage increases of 48%, 47%, and 33%, respectively. This uptick may be correlated with a 45% increase in the global population. Conversely, the overall prevalence, DALY rate, and incidence of urinary stones have been declining on a global scale over time. These findings suggest that while the number of patients has increased, the rate of new case emergence is decelerating. This trend can be attributed to major advances in diagnosis, prevention, and prognosis from the clinical experience accumulated by urological professionals with physicians over the past 32 years, as well as the development of advanced diagnostic and therapeutic technical systems. In 2021, low-SDI areas exhibited the lowest rates of prevalence, incidence, and DALYs, while medium-high SDI areas reported the highest figures. However, both prevalence and incidence in low-SDI regions increased by 150% from 1990 to 2021. This trend aligns with previous epidemiological transition models, which indicated a greater burden of urolithiasis in the 20–54 age group in countries with higher SDI levels [30]. Higher valuesof the Social Development Index (SDI) are often associatedwith stronger healthcare systems and higher-quality medical services, both of which lead to a lower disease burden. In this investigation, the highest prevalence, incidence, and disability-adjusted life years (DALYs) associated with urolithiasis in the 20–54 age group were observed in regions classified as moderately high SDI, while the most substantial rise in cases was documented in low SDI regions.This study implies that countries with moderate to high SDI frequently have regions with more sophisticated economies and better social infrastructure. Countries with enormous populations, such as China and India, have a proportionally big number of patients. Furthermore, as the economies of middle and high SDI regionsgrow, so does the number of reported causes and diagnoses of urolithiasis among those

aged 20–54.However, it is important to recognize that growing urbanization and industrialization have caused significant shifts in living choices. These alterations include increased sedentary behavior, higher professional stress, a lack of physical activity due to overexertion, and insufficient access to water or restroom facilities which impedes regular hydration [31]. Such conditions may cause decreased fluid intake and urine output, increasing the risk of stone formation and worsening the incidence of urolithiasis [32,33]. Certain vocations that require prolonged exposure to high temperatures, such as steelworkers, glass producers, and mechanics, have an increased risk of acquiring urinary stones [34]. In a study of Swedish battery industry workers, cumulative cadmium exposure was linked to an increase the risk of kidney stone formation [35]. A 7-year prospective analysis found that long-term cadmium exposure is related with an increased risk of upper urinary tract stones [36]. Furthermore, a case study revealed that the risk of kidney stones among drivers could be linked to tight workplace requirements surrounding toilet breaks [37]. The Korean study's postulated "healthy worker effect" is a significant problem in epidemiological studies on occupational health. Employed people had a higher prevalence and incidence of urinary stones than unemployed people [38]. The rise in the frequency, incidence, and disability-adjusted life years (DALYs) associated with urolithiasis among the 20–54 age group in low SDI nations is especially noticeable. This may be attributed to advancements in medical facilities, such as the widespread use of computers, driven by economic development and international support. Tomography (CT) and X-rays in clinical practice [39], as well as increased awareness of self-care in the population.As a result, urolithiasis detection rates have increased significantly. According to research, patients in wealthy locations and those with private insurance are more likely to receive imaging diagnosis, such as CT scans, during medical appointments. Patients from developing countries, on the other hand, are less likely to undergo any type of diagnostic imaging, such as KUB, ultrasound, or CT [40]. Urinary stone prevalence and incidence appear to be higher in developing nations with rapid socioeconomic growth. Although various elements may be involved, improvements in socioeconomic conditions are obviously important [25]. By encouraging environmental governance policies to lower the risk of heavy metal and mineral pollution, strengthening urban and rural infrastructure to ensure clean drinking water supply, improving the basic medical security system to cover stone screening and treatment, publicizing health education to raise awareness of disease prevention, improving the industrial structure to reduce agricultural and industrial pollution, and improving residents' employment and income to improve diet and medical access.

In terms of age and gender pattern, the prevalence and incidence of urinary stones in the global 20–54 age range increased steadily with age in 2021. Urinary stone prevalence and incidence were substantially lower in the 20–24 age group compared to the 50–54 age group.This trend may be related to the metabolism of calcium oxalate, the most prevalent component of kidney stones. However, there are significant regional differences: in Europe, the incidence of calcium oxalate stones rises between the ages of 40 and 50, whereas in North Africa, it peaks between 16 and 39 years [41]. In contrast, Wu et al. found that the peak age for calcium oxalate stones in southern China is between 19 and 40 years [42]. The researchers Yang et al., on the other hand, found that the peak prevalence occurred in eastern China between the ages of 30 and 50 [43]. Various ethnic groups, regions, and food habits all contribute to this. Hypertension and metabolic syndrome have both been linked to the development of kidney stones in southern Italian populations [44]. In contrast, in South Korea, the odds ratio for the presence of metabolic syndrome among 34,895 individuals undergoing general health screening was shown to be 1.25 in connection to the occurrence of kidney stones [45]. So it's more common in older people than in younger people in the same age group.

Males appear to be more susceptible to urolithiasis than females, with prior research indicating a male-to-female ratio ranging from 1.7:1–3:1 [46,47]. Furthermore, calcium oxalate and uric acid stones are more common observed in men than in women, which is consistent with current research in specific regions of China [42,48]. The actual pathophysiology driving the sex difference in urolithiasis is unknown, while various putative causes have been discovered. In terms of eating habits, males consume more alcohol and coffee and eat more meat than females [49]. Physiologically, testosterone can promote stone formation, but estrogen appears to decrease it via controlling the production of 1,25-dihydroxyvitamin D [50]. However, the conclusions of numerous investigations appear to disagree. For example, some studies emphasize

the inhibitory effect of estrogen on stone formation, while others imply that estrogen replacement therapy in postmenopausal women may be a risk factor for urolithiasis [51].

In summary, our study found that the global burden of urolithiasis among people aged 20–54 reduced between 1990 and 2021. However, given the importance of this age group for social productivity, urolithiasis is anticipated to continue causing large health and developmental costs, particularly among the 50–54 age group in middle-and high-SDI countries like China and India. It is critical to emphasize that urolithiasis itself carries significant health hazards. Urolithiasis patients had a 1.3 times higher chance of developing diabetes, 1.5 times higher risk of hypertension, twice the risk of metabolic syndrome, and a 2–4 times higher risk of cardiovascular disease [52]. These dangers necessitate additional preventive policies. Furthermore, while developing global health targets, disparities in the burden of urolithiasis across genders and age groups should be taken into account in order to develop more effective and suitable medical and health policies for urolithiasis prevention and early treatment.

The limitations of this study begin with the constraints of the GBD study itself. The primary data sources include censuses, household surveys, civil registration and vital statistics, satellite imaging,and so on. As a result, the quality of the data gathered in this study varied, and the predictive value of the modeling effort was obtained without the original data. Furthermore, the GBD study made no distinction between urolithiasis kinds, which may differ by geography. Stone compositio and location are not distinguished, which makes it impossible to guide prevention strategies, and the over-inclusion of asymptomatic stones overestimates the actual medical needs.As a result, the quality of the data gathered in this study varied, and the predictive value of the modeling effort was determined without the original data. Furthermore, gender differences may be influenced by clinic bias, and the "high burden" of SDI stratification may reflect high diagnostic rates rather than true incidence. As a result, more research is needed to explore the prevalence patterns of urolithiasis in different sections of the urogenital tract and with various chemical compositions. In the future, multi-center prospective cohort studies are needed to integrate stone composition analysis, environmental exposure monitoring and health economic evaluation to develop precise prevention and control strategies.

## Conclusions

In conclusion, since 1990, there has been an increase in urolithiasis cases, DALYs, and fatalities worldwide among those aged 20–54. While the frequency and DALY rates in low SDI countries grew at the quickest rate, those in high SDI countries were declining. Men had a greater prevalence rate and DALY rate than women did. Because of the enormous illness burden that urolithiasis causes, policies for particular age groups must be developed in order to prevent and treat urolithiasis, alleviate the social medical and health burden, and lower the working age group's quality of life and productivity.

## Supporting information

**S1 Table.  The incidence of urolithiasis cases and rates among 20-54 in 1990 and 2021, and the trends from 1990 to 2021.**
(XLSX)

**S2 Table.  The DALY of urolithiasis cases and rates among 20-54 in 1990 and 2021, and the trends from 1990 to 2021.**
(XLSX)

**S3 Table.  Percentage change in EAPC.**
(XLSX)

**S4 Table.  20-54 Structure of urolithiasis Prevalence Rate among Males and Females, 2021.**
(XLSX)

**S5 Table. 20-54 Structure of urolithiasis Incidence Rate among Males and Females, 2021.**
(XLSX)

**S6 Table. 20-54 Structure of urolithiasis DALYs Rate among Males and Females, 2021.**
(XLSX)

**S7 Table. 20-54 and Sex Structure of urolithiasis Prevalence, 1990-2021.**
(XLSX)

**S8 Table. 20-54 and Sex Structure of urolithiasis Incidence, 1990-2021.**
(XLSX)

**S9 Table. 20-54 and Sex Structure of urolithiasis DALYs, 1990-2021.**
(XLSX)

**S10 Table. Temporal Joinpoint Analysis of urolithiasis 20-54 Prevalence Rates, 1990-2021.**
(XLSX)

**S11 Table. Temporal Joinpoint Analysis of urolithiasis 20-54 Incidence Rates, 1990-2021.**
(XLSX)

**S12 Table. Temporal Joinpoint Analysis of urolithiasis 20-54 DALYs Rates, 1990-2021.**
(XLSX)

## Author contributions

**Conceptualization:** Weitao Yao.

**Data curation:** Weitao Yao, Qiang Jing, Fan Liu.

**Formal analysis:** Weitao Yao, Qiang Jing, Fan Liu.

**Investigation:** Weitao Yao, Xiaobin Yuan.

**Methodology:** Weitao Yao.

**Project administration:** Xuhui Zhang.

**Resources:** Xiaobin Yuan.

**Software:** Xin Wei, Xiaobin Yuan.

**Supervision:** Xuhui Zhang.

**Visualization:** Fan Liu.

**Writing – original draft:** Weitao Yao.

**Writing – review & editing:** Xuhui Zhang.

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
