## [Decision Letter · Decision Letter 0]

PONE-D-24-54570Urolithiasis panorama in working-age people: results from the analysis of Global Burden of Disease databasePLOS ONE

Dear Dr. Zhang,

Thank you for submitting your manuscript to PLOS ONE. After careful consideration, we feel that it has merit but does not fully meet PLOS ONE’s publication criteria as it currently stands. Therefore, we invite you to submit a revised version of the manuscript that addresses the points raised during the review process.

It is noteworthy that the first reviewer explicitly mentioned the use of AI during the review process. The use of AI may potentially affect the integrity of the peer-review process. The author is advised to focus on the content of the reviewer's comments and improve the manuscript accordingly. If the author disagrees with the content of the review, a polite and respectful tone should be maintained in the response.

We look forward to receiving your revised manuscript.

Kind regards,

Pengpeng Ye

Academic Editor

PLOS ONE

Journal Requirements:

3. Please note that your Data Availability Statement is currently missing the repository name and/or the DOI/accession number of each dataset OR a direct link to access each database. If your manuscript is accepted for publication, you will be asked to provide these details on a very short timeline. We therefore suggest that you provide this information now, though we will not hold up the peer review process if you are unable.

4. We note that Figure 4 in your submission contain map/satellite images which may be copyrighted. All PLOS content is published under the Creative Commons Attribution License (CC BY 4.0), which means that the manuscript, images, and Supporting Information files will be freely available online, and any third party is permitted to access, download, copy, distribute, and use these materials in any way, even commercially, with proper attribution. For these reasons, we cannot publish previously copyrighted maps or satellite images created using proprietary data, such as Google software (Google Maps, Street View, and Earth). For more information, see our copyright guidelines: http://journals.plos.org/plosone/s/licenses-and-copyright.

 a. You may seek permission from the original copyright holder of Figure 4 to publish the content specifically under the CC BY 4.0 license. 

Reviewers' comments:

Reviewer's Responses to Questions

**Comments to the Author**

1. Is the manuscript technically sound, and do the data support the conclusions?

Reviewer #1: Yes

Reviewer #2: Yes

Reviewer #3: Yes

Reviewer #4: Yes

Reviewer #5: Partly

Reviewer #6: Yes

2. Has the statistical analysis been performed appropriately and rigorously? 

Reviewer #1: I Don't Know

Reviewer #2: Yes

Reviewer #3: Yes

Reviewer #4: Yes

Reviewer #5: Yes

Reviewer #6: No

3. Have the authors made all data underlying the findings in their manuscript fully available?

Reviewer #1: No

Reviewer #2: Yes

Reviewer #3: Yes

Reviewer #4: Yes

Reviewer #5: Yes

Reviewer #6: Yes

4. Is the manuscript presented in an intelligible fashion and written in standard English?

Reviewer #1: Yes

Reviewer #2: Yes

Reviewer #3: Yes

Reviewer #4: Yes

Reviewer #5: Yes

Reviewer #6: No

5. Review Comments to the Author

Reviewer #1: Abstract

Weaknesses:

1. Overuse of technical terms: While the abstract includes key findings, it may not be easily understood by a broader audience due to terms like "EAPC," "SDI," and "DALYs" being introduced without explanation.

2. Lack of focus on implications: While findings are well-presented, the abstract could better emphasize the practical implications of the results (e.g., healthcare strategies or policy recommendations).

3. No mention of limitations: Abstracts should briefly mention significant limitations, especially for studies analyzing large datasets.

Introduction

1. Limited context: While the introduction describes urolithiasis prevalence, it doesn’t sufficiently address why the 20–54 age group was chosen as the focus or the gaps in prior studies.

2. Weak research justification: The rationale for the study is mentioned but not strongly emphasized.

3. Inconsistent citation style: Some claims lack proper citation, while others reference multiple sources for the same point.

Methods

Weaknesses:

1. Insufficient details: While the methods describe data sources and analysis techniques, they lack details about how the data was processed (e.g., criteria for excluding outliers or handling missing data).

2. Clarity of statistical methods: Terms like "joinpoint regression" and "EAPC" are introduced without explanation, making it difficult for readers unfamiliar with these concepts to follow.

3. Reproducibility issues: The section doesn’t mention whether code or analysis scripts are available for reproducibility.

Results

Weaknesses:

1. Dense presentation: Results are presented in a dense format with little interpretation, making it difficult for readers to extract key messages.

2. Limited focus on practical insights: While the numbers are well-detailed, there’s insufficient focus on what these trends mean in a practical context.

3. Underutilization of visuals: Figures and tables are used but could be more impactful if tailored to highlight key trends and findings.

Discussion

Weaknesses:

1. Lack of critical analysis: While results are discussed, there’s limited critical engagement with potential biases or alternative explanations for the findings.

2. Overgeneralization of results: Some conclusions, such as the need for "focused intervention techniques," are too broad without specifying actionable recommendations.

3. Insufficient limitations: The study acknowledges data limitations but doesn’t address other potential weaknesses, such as reliance on global datasets or differences in healthcare systems.

Reviewer #2: The study titled "Urolithiasis panorama in working-age people: results from the analysis of Global

Burden of Disease database" has a good subject. However, there are some comments that need to be addressed. Please see the attached file.

Reviewer #3: The manuscript is well written, just in figure 2 (below line 214) incidence rate in the table written in a wrong way.

The figures clarity should be adjusted.

Discussion: the aim and findings of the result was repeated in details in the discussion, instead it will be better to compare the result and justify the increase in prevalence in more details.

Reviewer #4: recommend to be accept with minor corrections

Review report

Review report of the manuscript titled ″Urolithiasis panorama in working-age people: results from the analysis of Global Burden of Disease database 1990-2021″ is below. Here are my suggestions:

Title

• The title is informative and reflects the study objective.

Abstract

• The ethical consideration is not specified.

Introduction

• The study rationale and objectives are clearly identified.

Methods

• The duration of the study is clearly specifying.

• Ethical considerations clearly identified.

Results

• The results highlight the study objectives.

Discussion

• The impact of the study on the literature is clarified.

• The findings are consistently discussed.

References

• The references updated and related to the study objectives.

Best regards

Reviewer #5: The manuscript titled "Urolithiasis panorama in working-age people: results from the analysis of the Global Burden of Disease database" presents a comprehensive analysis of urolithiasis trends from 1990 to 2021 among individuals aged 20–54. Using the Global Burden of Disease (GBD) database, the study examines prevalence, incidence, and Disability-Adjusted Life Years (DALYs) globally, across sociodemographic index (SDI) regions, and at national levels. The findings indicate a significant increase in the burden of urinary stones, particularly in middle SDI regions and among older individuals in the working-age group.

1. The title should indicate the focus on epidemiological trends (e.g., "Epidemiological Trends in Urolithiasis Among Working-Age Populations...").

2. Why was the age range of 20–54 chosen? Given that urolithiasis also affects individuals outside this range, a justification is necessary. Were trends in younger (<20) and older (>54) individuals considered?

3. The manuscript mentions joinpoint regression to detect trends. How were these parameters optimized?

4. While the study discusses trends, it implies potential causation (e.g., economic and healthcare improvements in high SDI regions). Were any confounders or alternative explanations considered?

Reviewer #6: I appreciate the efforts put into this paper and the valuable insights it provides. To further enhance the readability and accuracy of the manuscript, I offer the following recommendations:

1. Language and Terminology

- Sentence Structure: The clarity of several sentences needs improvement. Additionally, there are instances where the same sentence is repeated. Revising these sections for conciseness and coherence would be beneficial.

- Vocabulary: Some terms require more precise usage. For instance, the term "epidemic" is not always appropriately used; the author might have intended to indicate an "increase in cases." Similarly, the term "migraine" used in line 274 needs reconsideration for accuracy.

2. Tables and Figures

- Tables: Certain tables, such as Table 2, are excessively lengthy and may overwhelm the reader. Consider condensing these tables to present the most critical data.

- Figures: The quality of the figures needs enhancement. Many figures appear blurry and require higher resolution to ensure clarity and readability.

3. Data Analysis

- Use of Absolute Numbers: While absolute numbers are acceptable, they may not accurately reflect changes over time when comparing data. It is important to consider population size when analyzing trends. For example, 3 cases in a population of 100 are proportionally more significant than 5 cases in a population of 1000.

I hope these recommendations are taken into consideration to improve the manuscript and better serve the readership. Thank you once again for your hard work and dedication to this research.

6. PLOS authors have the option to publish the peer review history of their article (what does this mean? ). If published, this will include your full peer review and any attached files.

**Do you want your identity to be public for this peer review?** For information about this choice, including consent withdrawal, please see our Privacy Policy .

Reviewer #1: No

Reviewer #2: No

Reviewer #3: **Yes: ** Mawada Hassan Abdelmagied

Reviewer #4: No

Reviewer #5: No

Reviewer #6: **Yes: ** Mohamed Abdelbaqy

---

## [Author Response · Author response to Decision Letter 1]

8 May 2025

Response to Reviewers#1

Abstract

1. Overuse of technical terms: While the abstract includes key findings, it may not be easily understood by a broader audience due to terms like "EAPC," "SDI," and "DALYs" being introduced without explanation.

Re We appreciate your insightful feedback. Regarding your observation the that abstract introduces terms like "EAPC", "SDI" and "DALYs" without explanation, which might not be easily understood by a wider audience, we have explained them when we first mentioned in the following text. However, due to the limited number of words in the abstract, we may not be able to elaborate on them in the abstract.

2. Lack of focus on implications: While findings are well-presented, the abstract could better emphasize the practical implications of the results (e.g., healthcare strategies or policy recommendations).

Re Thank you for your review and feedback. In response to the questions you raised, I have added some practical significance of the research results in the abstract, which is in line 48-51.

3. No mention of limitations: Abstracts should briefly mention significant limitations, especially for studies analyzing large datasets.

Re Thank you for your outstanding review comments, I have suggested the limitations of this study in the abstract at lines 52-53.

Introduction

1. Limited context: While the introduction describes urolithiasis prevalence, it doesn’t sufficiently address why the 20–54 age group was chosen as the focus or the gaps in prior studies.

Re Thank you for your valuable advice. We have made additional comments on the following two aspects (the revised content has been marked in lines 83-91 of the manuscript).

2. Weak research justification: The rationale for the study is mentioned but not strongly emphasized.

Re Thank you for your valuable advice. We have made additional comments on the following two aspects (the revised content has been marked in lines 92-95 of the manuscript).

3. Inconsistent citation style: Some claims lack proper citation, while others reference multiple sources for the same point.

Re Thank you for your careful review of the rigor of literature citation. We fully agree that literature citation should follow the principle of necessity and have systematically optimized the citation.

Methods

1. Insufficient details: While the methods describe data sources and analysis techniques, they lack details about how the data was processed (e.g., criteria for excluding outliers or handling missing data).

Re�I thank the reviewer for his attention to my data processing methods. Regarding the data quality control standards, the following is added:All analyses and graphical presentation were performed with the use of statistical software R, version 3.5.2. Data GBD 2021, which provides the latest epidemiological data estimates of the burden of 371 diseases and injuries in 21 GBD areas and 204 countries and territories for the period 1990 to 2021. After comprehensive verification, no outliers were found in this data set, the completeness reached 100%, and no data missing occurred.

2. Clarity of statistical methods: Terms like "joinpoint regression" and "EAPC" are introduced without explanation, making it difficult for readers unfamiliar with these concepts to follow.

Re Thanks to the reviewers for their review comments, we have included short explanations after terms like "Joinpoint regression" and "EAPC" for better understanding by readers.

3. Reproducibility issues: The section doesn’t mention whether code or analysis scripts are available for reproducibility.

Re Thank you for your review. The following are partial codes used for the analysis of prevalence, incidence, DALYS, and the codes used for EAPC calculations.

library(dplyr)

library(ggplot2)

setwd("F:\\GBD\\20-54")

options(scipen = 200)

WCBA<-read.csv("worldIHME-GBD_2021_DATA-48862eba-1.csv")

region_order <- read.csv("order.csv",header = F)

WCBA$location <- factor(WCBA$location,

levels=region_order$V1,

ordered=TRUE)

#-------------------------Table1:患病率prevalence------------------------

##-------Number-----------

###----1990------

WCBA_1990<-subset(WCBA,

WCBA$year==1990 &

WCBA$age=='20-54 years' &

WCBA$metric=='Number' &

WCBA$measure=='DALYs (Disability-Adjusted Life Years)')

WCBA_1990<-WCBA_1990[,c(2,8,9,10)] #只需要选择需要的变量地区及对应的数值

WCBA_1990$val<-round(WCBA_1990$val/1,2)

WCBA_1990$upper<-round(WCBA_1990$upper/1,2)

WCBA_1990$lower<-round(WCBA_1990$lower/1,2)#取两位小数

WCBA_1990$Num_1990 <- paste0("(", WCBA_1990$lower, "-", WCBA_1990$upper, ")")

WCBA_1990$Num_1990 <- paste0(WCBA_1990$val, " ", WCBA_1990$Num_1990)

###----2021------

WCBA_2021<-subset(WCBA,

WCBA$year==2021 &

WCBA$age=='20-54 years' &

WCBA$metric=='Number' &

WCBA$measure=='DALYs (Disability-Adjusted Life Years)'

)

WCBA_2021<-WCBA_2021[,c(2,8,9,10)] #只需要选择需要的变量地区及对应的数值

WCBA_2021$val<-round(WCBA_2021$val/1,2)

WCBA_2021$upper<-round(WCBA_2021$upper/1,2)

WCBA_2021$lower<-round(WCBA_2021$lower/1,2)#取整

WCBA_2021$Num_2021 <- paste0("(", WCBA_2021$lower, "-", WCBA_2021$upper, ")")

WCBA_2021$Num_2021 <- paste0(WCBA_2021$val, " ", WCBA_2021$Num_2021)

###-----percentage change------

WCBA_1990to2021_Numchange <- cbind(WCBA_1990,WCBA_2021)

WCBA_1990to2021_Numchange$percentage_change_val <- round(((WCBA_2021$val - WCBA_1990$val)/WCBA_1990$val),2)

WCBA_1990to2021_Numchange$percentage_change_lower <- round(((WCBA_2021$lower - WCBA_1990$lower)/WCBA_1990$lower),2)

WCBA_1990to2021_Numchange$percentage_change_upper <- round(((WCBA_2021$upper - WCBA_1990$upper)/WCBA_1990$upper),2)

WCBA_1990to2021_Numchange$percentage_change <- paste0("(", WCBA_1990to2021_Numchange$percentage_change_lower, "-", WCBA_1990to2021_Numchange$percentage_change_upper, ")")

WCBA_1990to2021_Numchange$percentage_change <- paste0(WCBA_1990to2021_Numchange$percentage_change_val, " ", WCBA_1990to2021_Numchange$percentage_change)

#整理

WCBA_NUM_1990<- WCBA_1990[,c(1,5)]

WCBA_NUM_2021<- WCBA_2021[,c(1,5)]

WCBA_NUM_1990to2021 <-WCBA_1990to2021_Numchange[,c(1,11)]

WCBA_Prelavence_NUM_Table1 <-merge(WCBA_NUM_1990,WCBA_NUM_2021,by="location",all.x=T)

WCBA_Prelavence_NUM_Table1 <-merge(WCBA_Prelavence_NUM_Table1,WCBA_NUM_1990to2021,by="location",all.x=T)

##-------Rate-----------

###----1990------

WCBA_1990<-subset(WCBA,

WCBA$year==1990 &

WCBA$age=='20-54 years' &

WCBA$metric=='Rate' &

WCBA$measure=='DALYs (Disability-Adjusted Life Years)'

)

WCBA_1990<-WCBA_1990[,c(2,8,9,10)] #只需要选择需要的变量地区及对应的数值

WCBA_1990$val<-round(WCBA_1990$val,2)

WCBA_1990$upper<-round(WCBA_1990$upper,2)

WCBA_1990$lower<-round(WCBA_1990$lower,2)#取2位小数

WCBA_1990$Rate_1990 <- paste0("(", WCBA_1990$lower, "-", WCBA_1990$upper, ")")

WCBA_1990$Rate_1990 <- paste0(WCBA_1990$val, " ", WCBA_1990$Rate_1990)

###----2021------

WCBA_2021<-subset(WCBA,

WCBA$year==2021 &

WCBA$age=='20-54 years' &

WCBA$metric=='Rate' &

WCBA$measure=='DALYs (Disability-Adjusted Life Years)')

WCBA_2021<-WCBA_2021[,c(2,8,9,10)] #只需要选择需要的变量地区及对应的数值

WCBA_2021$val<-round(WCBA_2021$val,2)

WCBA_2021$upper<-round(WCBA_2021$upper,2)

WCBA_2021$lower<-round(WCBA_2021$lower,2)#取整

WCBA_2021$Rate_2021 <- paste0("(", WCBA_2021$lower, "-", WCBA_2021$upper, ")")

WCBA_2021$Rate_2021 <- paste0(WCBA_2021$val, " ", WCBA_2021$Rate_2021)

###-------EAPC--------

EAPC<-subset(WCBA,

WCBA$age=='20-54 years' &

WCBA$metric=='Rate' &

WCBA$measure=='Incidence')

EAPC <-EAPC[,c(2,7,8)]

country<-WCBA_1990$location

EAPC_cal <-data.frame(location = country,

EAPC = rep(0, times = 204),

UCI = rep(0, times = 204),

LCI = rep(0, times = 204))

for(i in 1:204){

country_cal <- as.character(EAPC_cal[i,1])

a <- subset(EAPC,EAPC$location==country_cal)

a$y <-log(a$val)

# 使用lm函数进行线性回归分析

regression_model <- lm(y ~ year, data = a)

# 计算EAPC值

eapc_value <- 100*(exp(summary(regression_model)[["coefficients"]][2,1])-1)

low_eapc <- 100*(exp(summary(regression_model)[["coefficients"]][2,1]-1.96*summary(regression_model)[["coefficients"]][2,2])-1)

high_eapc <- 100*(exp(summary(regression_model)[["coefficients"]][2,1]+1.96*summary(regression_model)[["coefficients"]][2,2])-1)

EAPC_cal[i,2] <-round(eapc_value,2)

EAPC_cal[i,3] <-round(high_eapc,2)

EAPC_cal[i,4] <-round(low_eapc,2)

}

EAPC_cal$EAPC_ci <- paste0("(", EAPC_cal$LCI, "-", EAPC_cal$UCI, ")")

EAPC_cal$EAPC_cal <- paste0(EAPC_cal$EAPC, " ", EAPC_cal$EAPC_ci)

#整理

WCBA_Rate_1990<- WCBA_1990[,c(1,5)]

WCBA_Rate_2021<- WCBA_2021[,c(1,5)]

WCBA_Rate_1990to2021_EAPC <-EAPC_cal[,c(1,6)]

WCBA_Prelavence_Rate_Table1 <-merge(WCBA_Rate_1990,WCBA_Rate_2021,by="location",all.x=T)

WCBA_Prelavence_Rate_Table1 <-merge(WCBA_Prelavence_Rate_Table1,WCBA_Rate_1990to2021_EAPC,by="location",all.x=T)

#合并

WCBA_Prelavence_Table1<-merge(WCBA_Prelavence_NUM_Table1,WCBA_Prelavence_Rate_Table1,by="location",all.x=T)

write.csv(WCBA_Prelavence_Table1,"F:\\GBD\\20-54\\1.csv")

Here are some of the codes used to analyze the prevalence, incidence, DALYs and changes of urinary stones from 1990 to 2021 by country worldwide:

library(ggmap)

library(maps)

library(dplyr)

setwd("F:\\GBD\\20-54\\worldmap")

WCBA <- read.csv("IHME-GBD_2021_DATA-48862eba-1.csv",header = T) ##??ȡ????

options(scipen = 200)

unique(WCBA$location)

#-------------------------??????revalence------------------------

###----2021------

case_2021<-subset(WCBA,WCBA$year==1990 &

WCBA$age=='20-54 years' &

WCBA$metric=='Rate' &

WCBA$measure=='Incidence'

)

case_2021$val<-case_2021$val/1

case_2021 <- case_2021[,c(2,8)]

names(case_2021) <- c('location','case_2021','upper','lower')

country_asr <- case_2021

country_asr$val <- (case_2021$case_2021) ### ??ȡ?????

#########################################################################################

#### map

worldData <- map_data('world')

worldData <-subset(worldData,region !='Antarctica')

unique(country_asr$location)

unique(worldData$region)

country_asr$location <- as.character(country_asr$location)

##write.csv(country_asr,"F:\\GBD\\20-54\\worldmap\\2021DALYs.csv")

###???´???==????????һ???

country_asr$location[country_asr$location == 'Bolivarian Republic of Venezuela'] = 'Venezuela'

country_asr$location[country_asr$location == 'Principality of Andorra'] = 'Andorra'

country_asr$location[country_asr$location == 'Republic of the Union of Myanmar'] = 'Myanmar'

country_asr$location[country_asr$location == 'Kingdom of Belgium'] = 'Belgium'

country_asr$location[country_asr$location == 'Russian Federation'] = 'Russia'

country_asr$location[country_asr$location == 'Republic of Paraguay'] = 'Paraguay'

country_asr$location[country_asr$location == 'Islamic Republic of Afghanistan'] = 'Afghanistan'

country_asr$location[country_asr$location == 'Federative Republic of Brazil'] = 'Brazil'

country_asr$location[country_asr$location == 'Republic of Italy'] = 'Italy'

#country_asr$location[country_asr$location == 'Bolivia (Plurinational State of)'] = 'Bolivia'

country_asr$location[country_asr$location == 'Bolivia (Plurinational State of)'] = 'Bolivia'

country_asr$location[country_asr$location == 'Kingdom of Bhutan'] = 'Bhutan'

country_asr$location[country_asr$location == 'Grand Duchy of Luxembourg'] = 'Luxembourg'

country_asr$location[country_asr$location == 'State of Libya'] = 'Libya'

country_asr$location[country_asr$location == 'Republic of the Philippines'] = 'Philippines'

country_asr$location[country_asr$location == 'Republic of Vanuatu'] = 'Vanuatu'

country_asr$location[country_asr$location == 'Togolese Republic'] = 'Togo'

country_asr$location[country_asr$location == 'Eastern Republic of Uruguay'] = 'Uruguay'

country_asr$location[country_asr$location == 'French Republic'] = 'France'

country_asr$location[country_asr$location == "People's Republic of Bangladesh"] = 'Bangladesh'

country_asr$location[country_asr$location == 'Republic of Tajikistan'] = 'Tajikistan'

country_asr$location[country_asr$location == 'Republic of Ecuador'] = 'Ecuador'

country_asr$location[country_asr$location == 'Republic of Korea'] = 'South Korea'

country_asr$location[country_asr$location == 'Commonwealth of Dominica'] = 'Dominica'

country_asr$location[country_asr$location == 'Republic of Poland'] = 'Poland'

country_asr$location[country_asr$location == 'Brunei Darussalam'] = 'Brunei'

country_asr$location[country_asr$location == 'Kingdom of Cambodia'] = 'Cambodia'

country_asr$location[country_asr$location == "Antigua and Barbuda"] = 'Lands Antigua'

a <- country_asr[country_asr$location == "Lands Antigua",]

a$location <- 'Barbuda'

country_asr <- rbind(country_asr,a)

country_asr$location[country_asr$location == 'Republic of Uzbekistan'] = 'Uzbekistan'

country_asr$location[country_asr$location == 'Republic of Chile'] = 'Chile'

country_asr$location[country_asr$location == 'Republic of Austria'] = 'Austria'

country_asr$location[country_asr$location == "Democratic People's Republic of Korea"] = 'North Korea'

country_asr$location[country_asr$location == 'Lebanese Republic'] = 'Lebanon'

country_asr$location[country_asr$location == 'Kingdom of Sweden'] = 'Sweden'

country_asr$location[country_asr$location == "Saint Kitts and Nevis"] = 'Saint Kitts'

a <- country_asr[country_asr$location == "Saint Kitts",]

a$location <- 'Nevis'

country_asr <- rbind(country_asr,a)

country_asr$location[country_asr$location == 'Republic of Colombia'] = 'Colombia'

country_asr$location[country_asr$location == 'Republic of Honduras'] = 'Honduras'

country_asr$location[country_asr$location == "Saint Vincent and the Grenadines"] = 'Saint Vincent'

a <- country_asr[country_asr$location == "Saint Vincent",]

a$location <- 'Grenadines'

country_asr <- rbind(country_asr,a)

country_asr$location[country_asr$location == "People's Republic of China"] = 'China'

country_asr$location[country_asr$location == 'Republic of Armenia'] = 'Armenia'

country_asr$location[country_asr$location == 'Republic of Singapore'] = 'Singapore'

country_asr$location[country_asr$location == 'Republic of Zimbabwe'] = 'Zimbabwe'

country_asr$location[country_asr$location == 'Independent State of Samoa'] = 'Samoa'

country_asr$location[country_asr$location == 'Kingdom of Bahrain'] = 'Bahrain'

country_asr$location[country_asr$location == 'Republic of Albania'] = 'Albania'

country_asr$location[country_asr$location == 'Republic of Belarus'] = 'Belarus'

country_asr$location[country_asr$location == "United Kingdom of Great Britain and Northern Ireland"] = 'UK'

country_asr$location[country_asr$location == 'Republic of Mozambique'] = 'Mozambique'

country_asr$location[country_asr$location == 'Republic of Bulgaria'] = 'Bulgaria'

country_asr$location[country_asr$location == 'Republic of Peru'] = 'Peru'

country_asr$location[country_asr$location == 'Republic of Fiji'] = 'Fiji'

country_asr$location[country_asr$location == 'Republic of Malta'] = 'Malta'

country_asr$location[country_asr$location == 'Gabonese Republic'] = 'Gabon'

country_asr$location[country_asr$location == 'Republic of Guinea'] = 'Guinea'

country_asr$location[country_asr$location == 'Republic of Liberia'] = 'Liberia'

country_asr$location[country_asr$location == 'Republic of Sudan'] = 'Sudan'

country_asr$location[country_asr$location == 'Kingdom of Morocco'] = 'Morocco'

country_asr$location[country_asr$location == 'Federal Republic of Germany'] = 'Germany'

country_asr$location[country_asr$location == 'Republic of Nicaragua'] = 'Nicaragua'

country_asr$location[country_asr$location == 'Trinidad and Tobago'] = 'Trinidad'

a <- country_asr[country_asr$location == "Trinidad",]

a$location <- 'Tobago'

country_asr <- rbind(country_asr,a)

country_asr$location[country_asr$location == 'Kingdom of Denmark'] = 'Denmark'

country_asr$location[country_as

---

## [Decision Letter · Decision Letter 1]

Epidemiological Trends of Urolithiasis in Working-Age Populations: Findings from the Global Burden of Disease Study 1990–2021

PONE-D-24-54570R1

Dear Dr. Zhang,

We’re pleased to inform you that your manuscript has been judged scientifically suitable for publication and will be formally accepted for publication once it meets all outstanding technical requirements.

Kind regards,

Pengpeng Ye

Academic Editor

PLOS ONE

Additional Editor Comments (optional):

Reviewers' comments:

Reviewer's Responses to Questions

**Comments to the Author**

1. If the authors have adequately addressed your comments raised in a previous round of review and you feel that this manuscript is now acceptable for publication, you may indicate that here to bypass the “Comments to the Author” section, enter your conflict of interest statement in the “Confidential to Editor” section, and submit your "Accept" recommendation.

Reviewer #2: All comments have been addressed

Reviewer #3: All comments have been addressed

Reviewer #4: All comments have been addressed

2. Is the manuscript technically sound, and do the data support the conclusions?

Reviewer #2: Yes

Reviewer #3: Yes

Reviewer #4: Yes

3. Has the statistical analysis been performed appropriately and rigorously? 

Reviewer #2: Yes

Reviewer #3: Yes

Reviewer #4: Yes

4. Have the authors made all data underlying the findings in their manuscript fully available?

Reviewer #2: Yes

Reviewer #3: Yes

Reviewer #4: Yes

5. Is the manuscript presented in an intelligible fashion and written in standard English?

Reviewer #2: Yes

Reviewer #3: Yes

Reviewer #4: Yes

6. Review Comments to the Author

Reviewer #2: Now, the manuscript is well-written, scientifically sound, and presents a valuable contribution to the field.

No further revisions are needed at this stage.

Reviewer #3: Thank you for your reply and addressing all comments clearly. from my side the article is ready for publication

Reviewer #4: All the comments have been addressed, deserve to be accept

ALL the required correction have been done in a perfect way

7. PLOS authors have the option to publish the peer review history of their article (what does this mean? ). If published, this will include your full peer review and any attached files.

**Do you want your identity to be public for this peer review?** For information about this choice, including consent withdrawal, please see our Privacy Policy .

Reviewer #2: **Yes: ** Omar Al-Mahmood

Reviewer #3: No

Reviewer #4: No

---

## [Editor Report · Acceptance letter]

PONE-D-24-54570R1

PLOS ONE

Dear Dr. Zhang,

I'm pleased to inform you that your manuscript has been deemed suitable for publication in PLOS ONE. Congratulations! Your manuscript is now being handed over to our production team.

Kind regards,

on behalf of

Dr. Pengpeng Ye

Academic Editor

PLOS ONE
